# PRIVACY-PRESERVING LLM INTERACTION WITH SO-CRATIC CHAIN-OF-THOUGHT REASONING AND HOMO-MORPHICALLY ENCRYPTED VECTOR DATABASES

## ABSTRACT

Large language models (LLMs) are increasingly used as personal assistants, accessing sensitive user data such as emails and medical records. Users currently face a trade-off: they can send private records to powerful but untrusted LLM providers, increasing their exposure risk, or they can run less powerful models locally on trusted devices. We bridge this gap: our **Socratic Chain-of-Thought Reasoning (Socratic-CoT)** first takes a user query and sends it to a powerful, untrusted LLM, which generates a chain-of-thought prompt and sub-queries without accessing user data. Next, we embed these sub-queries and perform encrypted semantic search using our **Homomorphically Encrypted Vector Database** over up to one million entries of a single user's private data. This represents a realistic scale of personal documents accumulated over years of digital activity. Finally, we feed the chain-of-thought prompt and the decrypted records to a local language model to generate the contextualized response. On the LoCoMo long-context QA benchmark, our hybrid framework—combining GPT-4o with a local Llama-3.2-1B model—outperforms using GPT-4o alone by up to 7.1 percentage points. This demonstrates a first step toward systems in which tasks are decomposed and split between untrusted strong LLMs and weak local ones, preserving user privacy. We will release all code and implementations publicly to facilitate transparency and future research.

## 1 INTRODUCTION

Large language models (LLMs) are becoming the default backend for personal assistants that handle emails and schedule meetings (Liu et al., 2023; Song et al., 2025). These assistants must integrate data from heterogeneous sources using Retrieval-Augmented Generation (RAG) (Lewis et al., 2020b). While forwarding user queries with retrieved data to powerful but untrusted LLMs can improve performance, it also raises privacy concerns by potentially exposing private records to LLM providers or to hackers (Zeng et al., 2024b; Jiang et al., 2024). Conversely, restricting these operations to local trusted devices degrades performance (Liu et al., 2025). This raises the question: *Can we perform LLM interactions on private data without privacy risks or significant performance degradation?*

Existing privacy-preserving methods, such as data minimization or scrubbing personally identifiable information (PII), often sacrifice data utility or provide limited privacy through superficial suppressions (Xin et al., 2025). To bridge the privacy-utility gap, we propose a hybrid framework that clearly delineates trusted and untrusted environments (left and right sides of Figure 1), ensuring private data either remains strictly within local boundaries or is securely encrypted when externally stored or searched. We integrate two novel components: (1) **Socratic Chain-of-Thought Reasoning (Socratic-CoT)**, which enables challenging queries to be offloaded to a powerful external language model; and (2) a **Homomorphically Encrypted Vector Database**, a cryptographic system that allows semantic search over encrypted records without ever decrypting them. Our framework enables users to leverage external compute resources and cloud storage while maintaining complete privacy.

**Stage 1**: When the user, Alice, poses a query (see Figure 1, top), Socratic-CoT elicits a detailed chain-of-thought prompt and sub-queries from a powerful external LLM. We provide only the generic query to the external LLM without exposing any private records. Rather than directly providing an answer, we prompt the powerful LLM to generate chain-of-thought for reasoning and sub-queries for retrieval—in this case, questions about medications and travel history. This approach allows the

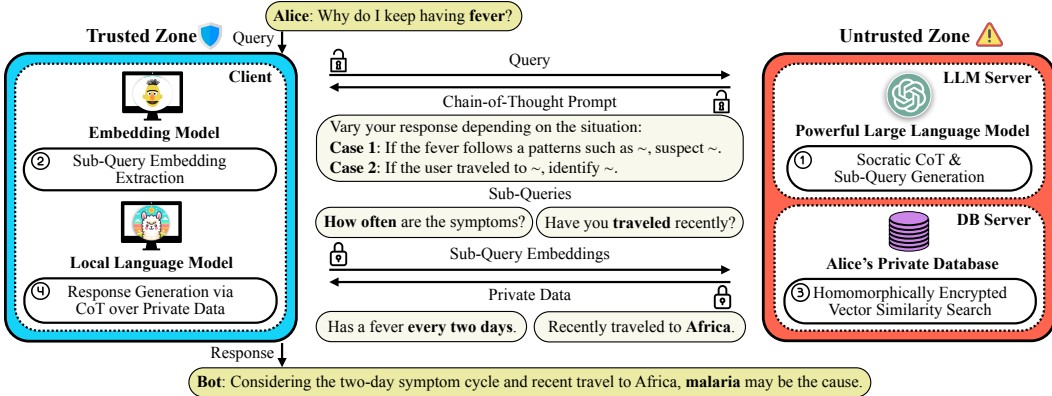

Figure 1: **Overview of our hybrid framework.** Upon receiving a query, a remote LLM generates a Chain-of-Thought (CoT) prompt and sub-queries (Stage 1) which are embedded locally (Stage 2), and used for our encrypted vector search on a remote database (Stage 3). Retrieved records are decrypted and provided with the CoT prompt as context to a local model to generate the final response (Stage 4).

powerful model to break down complex task into simpler ones, making it easier for the weaker local model to reason effectively when given access to private data. **Stage 2**: The sub-queries are then locally embedded for secure search over our encrypted database containing Alice's private records. **Stage 3**: Once the sub-query embeddings reach our Homomorphically Encrypted Vector Database, the system executes secure vector similarity search, where all key vectors are homomorphically encrypted and compared against a million encrypted key vectors. Our novel inner product algorithm computes similarity entirely in the encrypted domain in under one second using standard CPUs. The system then retrieves the corresponding encrypted records, returning top-$k$ matches from a million-entry store in encrypted format. **Stage 4**: Finally, a much smaller, weaker language model operating exclusively within the local trusted zone generates the final response, drawing on both the decrypted private records and the chain-of-thought prompt supplied by the stronger remote model.

We extensively evaluate our framework on two long-context QA benchmarks. *LoCoMo* assesses recall of extensive conversational histories (Maharana et al., 2024), while *MediQ* tests interactive medical consultation (Li et al., 2024b). We establish two baselines representing privacy extremes: (1) a (fully private) local-only baseline using Llama-3 with 1B, 3B, and 8B parameters, and (2) a (fully non-private) remote-only golden baseline using GPT-4o, Gemini-1.5-Pro, and Claude-3.5 Sonnet. Our approach provides a balanced trade-off between these extremes. Through our **Socratic-CoT**, the Llama 1B-parameter local model achieves an F1 score of 87.7 on LoCoMo, notably surpassing GPT-4o by 7.1 percentage points and the local-only baseline by 23.1 percentage points. This improvement likely stems from additional test-time computation enabled by the chain-of-thought process (Chen et al., 2024). For MediQ, the improvements are smaller due to domain-specific adaptation challenges, but they remain beneficial. Our **Homomorphically Encrypted Vector Database** efficiently searches entries from $10^6$ records in under one second on commodity CPUs, maintaining $> 99\%$ Recall@5 with a median storage overhead of just $\mathbf{5.8\times}$. *Collectively, our findings mark an important step toward privacy-preserving systems that effectively partition tasks between untrusted high-capacity LLMs and trusted lightweight local models, without requiring any additional post-training.*

## 2 PRIVACY-PRESERVING FRAMEWORK WITH SOCRATIC-COT

We present a privacy-preserving framework that enables powerful LLM reasoning while maintaining strict privacy guarantees, ensuring that sensitive user data is never exposed during interaction. This section outlines our approach, where Socratic Chain-of-Thought Reasoning (Socratic-CoT) is integrated into a framework that separates a trusted local zone and an untrusted cloud zone.

### 2.1 PROBLEM FORMULATION

**Background.** LLMs increasingly serve as personal assistants, processing sensitive user data such as emails and medical records (Zeng et al., 2024a; Qiu et al., 2024). Effective personal assistants require

two fundamental capabilities: (1) Context Retrieval: The model must determine which contextual data is necessary given the user query. As illustrated in Figure 1, a query such as *"Why do I keep having fever?"* may not provide sufficient information to personalize an answer. The model must generate targeted sub-queries to collect missing information, such as travel history that might reveal malaria risk factors (Lewis et al., 2020b). (2) Contextual Reasoning: The model must establish clear reasoning steps to accurately interpret the query in context. For instance, recognizing *a fever recurring every two days* in combination with *recent travel to Africa* strongly suggests *malaria*.

**Trade-off between Privacy and Performance.** While powerful cloud-based LLMs offer superior reasoning capabilities, they require users to expose private data to untrusted providers (Mireshghallah et al., 2023). Conversely, local models that preserve privacy lack the computational capacity for complex reasoning tasks. Local devices have limited computational resources insufficient for such reasoning, whereas powerful cloud LLMs cannot be trusted with sensitive data (Wang et al., 2024).

**Threat Model and Privacy Goal.** We protect against three adversaries: (1) the LLM provider that receives user queries, (2) the database provider storing encrypted records (Bonnetain et al., 2019), and (3) external attackers who may compromise these services (Hutchins et al., 2011). Even with standard encryption, providers often hold decryption keys, creating potential privacy risks through insider threats or security breaches (Cappelli et al., 2012; Hunker & Probst, 2011). User data must remain encrypted outside the local environment, with decryption keys never leaving the user's control. We assume that the local environment is secure, and that the server and network are semi-honest. Threats include network sniffing, server compromise, and insider analysis of ciphertexts or access patterns. Malicious queriers are excluded, since each user queries only their own authenticated database. The system must support complex reasoning and efficient retrieval while ensuring that untrusted components cannot access plaintext private data (Gentry, 2009b; Rivest et al., 1978).

## 2.2 OVERALL PROCESS

Figure 1 illustrates our framework's overall process, which separates computation into trusted and untrusted zones to balance privacy and performance. In the trusted zone (left side of the figure), the user's local device hosts a lightweight language model and embedding model with exclusive access to decryption keys, ensuring that sensitive data never leaves the user's control in plaintext form. The untrusted zone (right side of the figure) comprises cloud providers hosting: (1) a powerful LLM for abstract reasoning, and (2) an encrypted vector database storing the user's private records using homomorphic encryption (Gentry, 2009b; Brakerski et al., 2014), allowing secure processing without data decryption. Consider the medical consultation example: when a user asks "Why do I keep having fever?", the query flows to the remote LLM without exposing any private medical history. The powerful model generates targeted sub-queries (e.g., symptom frequency, travel history) that guide retrieval from the encrypted database, where private records remain protected even during search operations thanks to homomorphic encryption. We term this reasoning collaboration Socratic-CoT, where the remote LLM questions the local model in a Socratic manner without accessing private data.

## 2.3 SOCRATIC-COT OPERATIONS

Let $\mathcal{V}$ be the set of tokens and define $k$-tuples of $\mathcal{V}$ as

$$\mathcal{V}^k = \{(v_0, \ldots, v_{k-1}) \mid v_0, \ldots, v_{k-1} \in \mathcal{V}\},$$

with $\mathcal{V}^* = \bigcup_{k=0}^{\infty} \mathcal{V}^k$ denoting the set of all finite-length sequences.

We denote the CoT generator as $G_c$, the sub-query generator as $G_q$, the retriever as $R$, and the database as $\mathcal{D}$. Given a user query $x$, the remote LLM generates a CoT prompt $c$ and a set of sub-queries $s$. The client then embeds $s$, encrypts the embeddings, retrieves encrypted records $v$ from $\mathcal{D}$ via $R$, and finally the local model $L$ produces the response $y$, conditioned on $c$, $v$, and the conversation history $h$, where $x, y, c, s, v, h \in \mathcal{V}^*$:

$$c = G_c(x), \quad s = G_q(x), \quad v = R(s, \mathcal{D}), \quad y = L(x, c, v, h).$$

This design ensures that $G_c$ and $G_q$ operate only on the user's current query $x$, while private records in $\mathcal{D}$ remain encrypted and are processed exclusively by $L$ within the trusted local environment. Importantly, the chat history $h$ is never sent to the remote LLM; it is retained locally to support coherent multi-turn conversations. As a result, our framework preserves user privacy while enabling context-aware dialogue across multiple turns. Qualitative examples are provided in Appendix G.

## 3 Homomorphically Encrypted Vector Database

### 3.1 Motivations and Setup

The performance of personal assistants significantly improves when relevant user data is provided. Thus, seamless accumulation and integration of private data are crucial for powerful assistants. While storing data locally on the user's device (e.g., smartphones and laptops) allows strong security and quick retrieval, edge devices are inherently limited in disk storage. For example, storing a user's 1M records with embeddings requires about 103 GB, over 80% of a base 128 GB iPhone's capacity, as calculated in Appendix B. Consequently, leveraging cloud solutions becomes essential, offering extensive scalability. Moreover, cloud solutions offer various advantages, including multi-device synchronization, snapshot backups, and simplified management. Therefore, we present an encrypted vector database that combines the security of local storage with the scalability of the cloud.

The retrieval process within a vector database typically involves two sub-processes: search and return. The search phase computes similarity scores between a query vector and the key vectors stored in the database, and selects the top-$k$ entries. In the return phase, corresponding data values are fetched from the database based on the selected entry IDs. To ensure robust security, three main parts must be executed in an oblivious manner: inner product (IP) computations, top-$k$ selection, and data access. Homomorphic Encryption (HE) is particularly effective for inner product calculations, as it significantly reduces both communication rounds and client-side computation. However, top-$k$ selection, which involves numerous logical comparisons, becomes computationally intensive when directly implemented with HE (Hong et al., 2021). Therefore, we adopt a client-aided approach, enabling the client to efficiently select the top-$k$ entry IDs without excessive computational overhead. Finally, once the client identifies the relevant entry IDs, records are securely retrieved using Private Information Retrieval (PIR) protocols, which fetch values corresponding to these IDs without revealing which database entries are accessed. Existing sublinear PIR schemes that rely on preprocessing are impractical for dynamic databases due to high preprocessing costs (Li et al., 2023). To mitigate this issue, several lines of work (Menon & Wu, 2022; Mughees et al., 2021) propose single-server PIR protocols that operate efficiently without preprocessing, thereby enabling dynamic and rapid updates alongside secure and efficient searching. In this way, we can achieve the same security guarantees as the naive baseline, which is fully secure but significantly inefficient, as described in Appendix C.4. For clarity, a detailed illustration of the retrieval process is presented in Appendix C.5.

### 3.2 Secure Inner Product

We introduce a novel HE-based IP algorithm, specifically designed for vector databases with considerations for dynamic updates, data integrity, fast search, and efficient storage. The most significant difference between *semantic search* and a *vector database* is that the database must be dynamic, supporting insertion and deletion. An important observation is that HE operations should ideally not be used for insertion and deletion, as they accumulate errors and eventually corrupt the message.[1] Many existing HE-based IP algorithms are unsuitable for scenarios requiring dynamic updates. Current solutions for *encrypted semantic search* with a public database, such as Wally (Asi et al., 2024) and HERS (Engelsma et al., 2022), typically precompute key vectors in plaintext domain for fast search. However, this plaintext precomputation restricts dynamic updates in the ciphertext domain. In HERS, for instance, each key data point is distributed across different ciphertexts, necessitating complex HE operations for inserting or deleting keys along with their approximate values. This process can degrade data integrity over time due to accumulated errors resulting from frequent HE computations. One way to avoid HE computations during insertion and deletion is to assign one ciphertext per key, allowing insertion and deletion by simply appending or removing ciphertexts. Building on this insight, we designed a dedicated HE-based IP scheme for vector databases that enables exact updates.[2] The search process begins by computing the inner product between the query and the stored key vectors. Let us break down each step to derive the complete algorithm. The detailed mathematical notations used in the following derivations are provided in Appendix C.1.

---

[1] One may consider using bootstrapping (Gentry, 2009a) to clean the errors, but it is almost infeasible due to its high computational cost.

[2] CHAM (Ren et al., 2023) also supports exact updates but is far less efficient than ours (as shown in Section 4.2), incurring impractical latency and storage costs for real-world deployment.

**Inner Product.** Let the query vector be $\mathbf{q} = [\xi_i]_{0 \le i < r} \in \mathbb{R}^r$ and the key vector be $\mathbf{k} = [\kappa_i]_{0 \le i < r} \in \mathbb{R}^r$. The corresponding plaintext polynomials are encoded as:

$$\mathrm{q}(X) = \sum_{i=0}^{r-1} q_i \cdot X^{-si} = \sum_{i=0}^{r-1} \lfloor \Delta \cdot \xi_i \rceil \cdot X^{-si} \in \mathcal{R}_{*,d} \tag{1}$$

$$\mathrm{k}(X) = \sum_{i=0}^{r-1} k_i \cdot X^{si} = \sum_{i=0}^{r-1} \lfloor \Delta \cdot \kappa_i \rceil \cdot X^{si} \in \mathcal{R}_{*,d} \tag{2}$$

where $\Delta > 0$ is a scaling factor and $d = rs$. Here, the inner product $\langle \mathbf{q}, \mathbf{k} \rangle$ can be derived as:

$$\frac{1}{\Delta^2} \cdot (\mathrm{q} \cdot \mathrm{k})[0] \simeq \langle \mathbf{q}, \mathbf{k} \rangle \tag{3}$$

We denote the (scaled) score $\sigma$ as $\sigma = (\mathrm{q} \cdot \mathrm{k})[0]$. To pack multiple scores in a ciphertext for reducing communication, we extract the constant term from the ciphertext. We slightly modify the conventional homomorphic trace and write as:

$$\sum_{i=0}^{r-1} \varphi_i(\mathrm{q} \cdot \mathrm{k}) = r \cdot \sigma \tag{4}$$

where $\varphi_i = \mathrm{p}(X) \mapsto \mathrm{p}(X^{2i+1})$ is an automorphism over $\mathcal{R}_{*,d}$ for each $0 \le i < r$.

**Batching.** We pack $d$ scores $\sigma_0, \sigma_1, \ldots, \sigma_{d-1}$ into a single ciphertext. By Equation 4,

$$r \cdot \sum_{j=0}^{d-1} \sigma_j X^j = \sum_{j=0}^{d-1} \sum_{i=0}^{r-1} \varphi_i(\mathrm{q} \cdot \mathrm{k}_j) X^j = \sum_{i=0}^{r-1} \left[ \varphi_i(\mathrm{q}) \cdot \left( \sum_{j=0}^{d-1} \varphi_i(\mathrm{k}_j) X^j \right) \right] \tag{5}$$

where $\sigma_j = (\mathrm{q} \cdot \mathrm{k}_j)[0]$ for each $0 \le j < d$. The last term can be interpreted as an inner product between $(\varphi_i)_i$ and $\left( \sum_{j=0}^{d-1} \varphi_i(\mathrm{k}_j) X^j \right)_i$, separating query and key operations. The number of automorphisms for the query is independent of $n$ (when $n \ge d$), and we can precompute the keys.

**Caching.** The key observation is that from Equation 5,

$$\sum_{j=0}^{d-1} \varphi_i(\mathrm{k}_j) X^j = \varphi_i \left( \sum_{j=0}^{d-1} \mathrm{k}_j X^{j \cdot \mathtt{inv}(i)} \right) \tag{6}$$

where $\mathtt{inv}(i) = (2i+1)^{-1} \bmod 2d$ so that $\varphi_i(X^{\mathtt{inv}(i)}) = X$. This formula allows us to compute the automorphism $\varphi_i$ only once. Therefore, we can significantly reduce the number of (homomorphic) automorphisms from $d \log(r)$ to $r - 1$.

**Butterfly Decomposition.** For $\tilde{\mathbf{k}} = \left( \tilde{k}_j \right)_{0 \le j < d} \in \mathcal{R}_{q,d}^d$ and $\mathbf{k} = \left( \sum_{j=0}^{d-1} \tilde{k}_j X^{j \cdot \mathtt{inv}(i)} \right)_{0 \le i < r} \in \mathcal{R}_{q,d}^r$, let

$$\mathbf{M} = P \cdot \begin{bmatrix} X^0 & X^1 & X^2 & \cdots & X^{(d-1)} \\ X^0 & X^3 & X^6 & \cdots & X^{3(d-1)} \\ X^0 & X^5 & X^{10} & \cdots & X^{5(d-1)} \\ \vdots & \vdots & \vdots & \ddots & \vdots \\ X^0 & X^{2r-1} & X^{2(2r-1)} & \cdots & X^{(2r-1)(d-1)\}} \end{bmatrix} \in \mathcal{R}_{q,d}^{r \times d}$$

where $P \in \mathcal{R}_{q,d}^{r \times r}$ is a permutation matrix that corresponds to the permutation $i \mapsto \frac{(2i+1)^{-1}-1}{2} \bmod r : \{0, 1, \ldots, r-1\} \to \{0, 1, \ldots, r-1\}$. Then $\mathbf{k} = \mathbf{M}\tilde{\mathbf{k}}$ holds. Multiplying $\mathbf{M}$ to $\tilde{\mathbf{k}}$ requires $r(r-1)$ polynomial additions, which is not negligible. Therefore, we use a DFT-style butterfly decomposition to reduce the computational cost.

Define $\mathbf{k}' \in \mathcal{R}_{q,d}^r$ as follows: $\mathbf{k}'[i] = \sum_{j=0}^{s-1} \tilde{k}_{j+si} X^j$ for $0 \le i < r$.

Then for $\mathbf{k}'' = \mathbf{B}\mathbf{k} \in \mathcal{R}_{q,d}^r$, $\varphi_{i,r}(\mathbf{k}''[i]) = \varphi_i(\mathbf{k}[i])$ holds for $0 \leq i < r$, where

$$\mathbf{B} = P \cdot \begin{bmatrix} X^0 & X^s & X^{2s} & \cdots & X^{(d-s)} \\ X^0 & X^{3s} & X^{6s} & \cdots & X^{3(d-s)} \\ X^0 & X^{5s} & X^{10s} & \cdots & X^{5(d-s)} \\ \vdots & \vdots & \vdots & \ddots & \vdots \\ X^0 & X^{s(2r-1)} & X^{2s(2r-1)} & \cdots & X^{(2r-1)(d-s)\}} \end{bmatrix} \in \mathcal{R}_{q,d}^{r \times r} \tag{7}$$

and $\varphi_{i,r} : \mathcal{R}_{q,d} \to \mathcal{R}_{q,d}$ is a permutation on the coefficients that satisfies

$$\varphi_{i,r}(\mathrm{p})[(2i+1) \cdot s \cdot u + j] = \mathrm{p}[s \cdot u + j] \tag{8}$$

for $0 \leq j < s-1$ and $0 \leq u < r$. By leveraging the butterfly matrix decomposition, we reduce the number of polynomial additions to $r \log(r)$. The detailed algorithm is written in Algorithm 7.

**Removing the Leading Term $r$.** To remove the leading term $r$ from the result $r \cdot \sum_{j=0}^{d-1} \sigma_j X^j$, we multiply $r^{-1} \pmod{q}$ before automorphisms.

$$r \cdot \sum_{i=0}^{r-1} \left( \varphi_i(r^{-1} \cdot \mathrm{q}) \cdot \left[ r \cdot \varphi_i \left( \sum_{j=0}^{d-1} r^{-1} \cdot \mathrm{k}_j X^{j \cdot \mathtt{inv}(i)} \right) \right] \right) = r \cdot \sum_{j=0}^{d-1} \sigma_j X^j \tag{9}$$

Therefore, $\sum_{i=0}^{r-1} \left( \varphi_i(r^{-1} \cdot \mathrm{q}) \cdot \left[ r \cdot \varphi_i \left( \sum_{j=0}^{d-1} r^{-1} \cdot \mathrm{k}_j X^{j \cdot \mathtt{inv}(i)} \right) \right] \right) = \sum_{j=0}^{d-1} \sigma_j X^j$.

**Optimizations.** To further enhance the performance of the encrypted database, we incorporate several advanced techniques to optimize computation, storage, and accuracy. One key optimization is caching via key-query decoupling, which allows keys to be precomputed and cached independently of queries. This significantly reduces query response time by accelerating inner product computation. We also apply hoisting (Halevi & Shoup, 2018; Bossuat et al., 2021) to efficiently decompose queries, minimizing computational overhead. This technique, when combined with MLWE (Bae et al., 2023) and seed-based ciphertext generation, enables compact storage and efficient updates. Storage and update efficiency is further improved through batch processing and MLWE-based seeding strategies (Bos et al., 2018), which reduce ciphertext size and update costs. Finally, we improve numerical precision by removing leading constant terms (Chen et al., 2021) in homomorphic computations, resulting in more accurate query results. See Appendix C for more detailed explanation.

### 3.3 DATABASE OPERATIONS

See Table 1 for our database's API, enabling efficient search and dynamic updates with $O(1)$ complexity. We denote the client as **Alice (A)** and the server as **Bob (B)**. Public parameters `pp` are shared between them. The function `GenSK` generates secret keys used in Homomorphic Encryption (HE), Advanced Encryption Standard (AES), and Private Information Retrieval (PIR), while `GenSwk` produces the corresponding public keys of HE and PIR including switching keys for homomorphic operations. The finite-length sequences, such as textual queries $q$ and records $v$, are embedded using an encoder $E$. The vector database $\mathcal{D}$ maintains the following attributes: `num` (number of entries), `key` (stored key vectors), `value` (encrypted records), and `cache` (cached key vectors for efficient search). Encryption and decryption are performed using `EncryptHE`, `DecryptHE`, `EncryptAES`, and `DecryptAES`. The `Score` function computes similarity scores over encrypted vectors, and `TopK` selects the top-$k$ most relevant entries using a heap-based algorithm with $O(n \log k)$ complexity. Retrieved values are fetched securely using PIR protocols. To support dynamic updates, we include auxiliary operations such as `len` (entry count), `Append` (inserting new entries), `Switch` (deleting entries by overwriting them with the last entry), and `ReCache` (refreshing the cached key vectors). These operations are executed in a batched manner and achieve constant-time complexity.

### 3.4 SECURITY GUARANTEES.

Key vectors are encrypted using CKKS (Cheon et al., 2017). Values are encrypted using non-deterministic AES-256. The combination of HE and AES provides robust security of our vector database. That is, our database provides 128-bit IND-CPA security (Rogaway, 2004; Bossuat et al., 2024) and is quantum-resistant (Bonnetain et al., 2019; Micciancio & Regev, 2009).

**Algorithm 1** `Init`

**Require:** public parameters pp
1: **A:** sk ← GenSK(pp)
2: **A:** pk ← GenPK(pp)
3: **A:** Send pk to **Bob** =0

**Algorithm 2** `Search`

**Require:** query $q$, database $\mathcal{D}$
1: **A:** q ← $E(q)$
2: **A:** q ← EncryptHE(q)
3: **A:** Send q to **Bob**
4: **B:** s ← Score(q, $\mathcal{D}_{\texttt{cache}}$)
5: **B:** Send s to **Alice**
6: **A:** s ← DecryptHE(s)
7: **A:** $\mathcal{I}$ ← TopK(**s**) =0

**Algorithm 3** `Return`

**Require:** record ids $\mathcal{I}$
1: **A&B:** $\{v\}$ ← PIR($\mathcal{D}_{\texttt{value}}$, $\mathcal{I}$)
2: **A:** $\{v\}$ ← {DecryptAES($v$)} =0

**Algorithm 4** `Insert`

**Require:** set of records $\{v\}$, database $\mathcal{D}$
1: **A:** k ← $E(v)$
2: **A:** k ← EncryptHE(k)
3: **A:** $v$ ← EncryptAES($v$)
4: **A:** Send $\{(k, v)\}$ to **Bob**
5: **B:** $\mathcal{D}_{\texttt{num}}$ ← $\mathcal{D}_{\texttt{num}}$ + len($\{(k, v)\}$)
6: **B:** $\mathcal{D}_{\texttt{key}}$ ← Append($\mathcal{D}_{\texttt{key}}$, $\{k\}$)
7: **B:** $\mathcal{D}_{\texttt{value}}$ ← Append($\mathcal{D}_{\texttt{value}}$, $\{v\}$)
8: **B:** $\mathcal{D}_{\texttt{cache}}$ ← ReCache($\mathcal{D}_{\texttt{cache}}$, $\mathcal{D}_{\texttt{key}}$, $\{k\}$) =0

**Algorithm 5** `Delete`

**Require:** record ids $\mathcal{A}$, database $\mathcal{D}$
1: **A:** Send $\mathcal{A}$ to **Bob**
2: **B:** $\mathcal{D}_{\texttt{num}}$ ← $\mathcal{D}_{\texttt{num}}$ − len($\mathcal{A}$)
3: **B:** $\mathcal{D}_{\texttt{key}}$ ← Switch($\mathcal{D}_{\texttt{key}}$, $\mathcal{A}$)
4: **B:** $\mathcal{D}_{\texttt{value}}$ ← Switch($\mathcal{D}_{\texttt{value}}$, $\mathcal{A}$)
5: **B:** $\mathcal{D}_{\texttt{cache}}$ ← ReCache($\mathcal{D}_{\texttt{cache}}$, $\mathcal{D}_{\texttt{key}}$, $\mathcal{A}$) =0

Table 1: Set of algorithms for homomorphically encrypted vector database operations.

## 4 EXPERIMENTS

In this section, we empirically demonstrate the practicality of our approach. We first present the overall performance of our privacy-preserving framework, followed by ablations on Socratic-CoT. We then examine the accuracy of our homomorphically encrypted vector database and assess its latency and storage cost. Details of the experimental setup are provided in Appendix D.

### 4.1 PRIVACY-PRESERVING FRAMEWORK WITH SOCRATIC-COT

**Our framework improves local-only baselines by up to +27.6 percentage points.** Experiments are conducted on two question-answering benchmarks: LoCoMo (Maharana et al., 2024), for personal assistant scenarios, and MediQ (Li et al., 2024a), for medical consultation scenarios. Both benchmarks require retrieving relevant private records and performing complex reasoning to predict an accurate answer. Consequently, these experiments assess the model's ability to retrieve appropriate private data for a given user query and generate well-contextualized responses accordingly. As shown in Table 2, our framework consistently outperforms the local-only baselines on both LoCoMo and MediQ. Delegating complex reasoning to powerful remote models yields substantial performance gains. Specifically, our hybrid framework improve over the local models by up to 23.1 percent points on LoCoMo and 27.6 on MediQ. These gains are especially notable in challenging scenarios requiring domain expertise, such as medical consultations. Despite operating under strict privacy constraints, our framework approaches—and in some cases surpasses—the performance of oracle baselines that operate without privacy constraints. This demonstrates the effectiveness of our approach in balancing strong privacy with high utility. To further analyze the detailed sources of these performance improvements, we conduct an ablation study on Socratic-CoT, which is presented in Appendix F.

### 4.2 HOMOMORPHICALLY ENCRYPTED VECTOR DATABASE

**Encrypted search retains > 99% accuracy.** We evaluate the search accuracy of our encrypted vector database using the LoCoMo (Maharana et al., 2024), Deep1B (Babenko & Lempitsky, 2016), and LAION (Schuhmann et al., 2022) datasets, which cover a wide range of modalities and vector dimensions. As noted in prior work (Cheon et al., 2017), homomorphic encryption is commonly known to introduce accuracy degradation as a result of error accumulation. Nevertheless, Table 3 shows that our database preserves high search fidelity across both plaintext-ciphertext and ciphertext-

| Baseline | Model | LoCoMo | MediQ |
|---|---|---|---|
| **Remote-Only Baselines** (oracle) | R1 GPT-4o | 80.6 | **81.8** |
| | R2 Gemini-1.5-Pro | 84.2 | 69.8 |
| | R3 Claude-3.5-Sonnet | **89.8** | 79.3 |
| **Local-Only Baselines** | L1 Llama-3.2-1B | 64.6 | 32.1 |
| | L2 Llama-3.2-3B | 68.7 | 43.2 |
| | L3 Llama-3.1-8B | 68.8 | 47.5 |
| **Hybrid Framework w/ Socratic-CoT** (ours) | L1 + R1 | 87.7 | 59.7 |
| | L1 + R2 | 85.1 | 49.7 |
| | L1 + R3 | 84.3 | 58.0 |
| | L2 + R1 | 85.9 | 60.7 |
| | L2 + R2 | 79.8 | 52.9 |
| | L2 + R3 | 74.6 | 59.0 |
| | L3 + R1 | 87.9 | 59.5 |
| | L3 + R2 | 88.0 | 52.1 |
| | L3 + R3 | 86.1 | 59.6 |

Table 2: Benchmark results on the **LoCoMo** and **MediQ** datasets. LoCoMo is evaluated by F1 score, while MediQ is evaluated by exact match. *Takeaway: Our hybrid framework significantly outperforms local-only baselines and approaches the oracle baselines without privacy constraints.*

ciphertext inner product computations. The encrypted database achieves accuracy comparable to its unencrypted counterpart, with both mean and maximum inner product errors remaining minimal. Furthermore, metrics such as 1-Recall@1, 1-Recall@5, and MRR@10 confirm that the top-k results from the encrypted database closely match those of the plaintext database. These results demonstrate that encrypted semantic search can be performed with negligible impact on accuracy.

| Dataset | Max Error | Mean Error | Std Error | MRR@10 | 1-Recall@1 | 1-Recall@5 |
|---|---|---|---|---|---|---|
| **Plaintext-Ciphertext** | | | | | | |
| **LoCoMo** | 6.94e-3 | 5.25e-3 | 4.53e-4 | 100 | 100 | 100 |
| **Deep1B** | 5.29e-5 | 6.42e-6 | 7.00e-11 | 99.97 | 99.96 | 99.99 |
| **LAION** | 1.06e-4 | 9.83e-6 | 1.36e-10 | 99.86 | 99.79 | 99.95 |
| **Ciphertext-Ciphertext** | | | | | | |
| **LoCoMo** | 1.61e-0 | 2.87e-1 | 2.15e-1 | 97.86 | 100 | 100 |
| **Deep1B** | 1.39e-3 | 1.71e-4 | 4.61e-8 | 99.59 | 99.21 | 99.97 |
| **LAION** | 2.70e-3 | 3.44e-4 | 1.87e-7 | 99.85 | 99.78 | 99.95 |

Table 3: Search accuracy across **LoCoMo**, **Deep1B**, and **LAION** datasets, evaluated under two settings: when the query is in plaintext (top) and when the query is ciphertext (bottom), with ciphertext keys in both cases. *Takeaway: Our encrypted vector database preserves high semantic search accuracy, achieving near-parity with the fully plaintext setting (both query and key).*

**Encrypted search scales to 1M entries with < 1 second latency.** A well-known challenge of homomorphic encryption is its heavy computational overhead. Nonetheless, our database achieves practical latency for large-scale vector similarity search. Figure 2 shows search latencies on the Deep1B (Babenko & Lempitsky, 2016) dataset. By leveraging efficient SIMD-style operations and low-precision arithmetic in the ciphertext space, the database scales linearly across sizes ranging from 1,000 to 1 million entries. At the million scale, end-to-end latency remains under one second—including encryption, computation, and communication—even under a slow network. This performance makes the encrypted database viable for real-time applications.

Our system also provides significant improvements over previous homomorphic encryption methods. As shown in Table 4, our approach consistently delivers faster runtimes than CHAM (Ren et al., 2023),

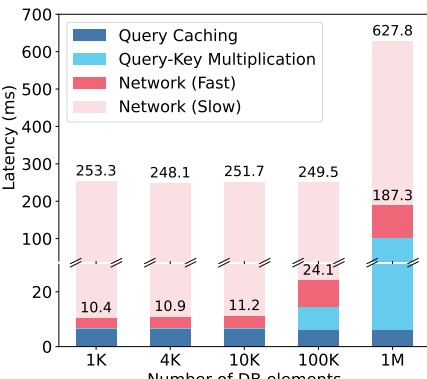

Figure 2: Search latency breakdown on the Deep1B dataset as the number of database elements increases. The results are obtained using 64 threads. Red and pink bars represent network communication time on fast and slow networks, respectively, with the numbers above each bar indicating the corresponding latency. Blue bars denote query caching time, while light-blue bars show query–key multiplication time. *Takeaway: Our encrypted search scales to 1M entries with < 1 second latency.*

an encrypted matrix–vector multiplication method designed for frequent updates. The performance gap widens with scale: while CHAM requires 84,543 ms to process 1 million entries, our method completes the same operation in just 2,280 ms—achieving a **37×** speed-up. This efficiency primarily stems from our query caching strategy, explained in Section 3.2, which restructures the key-switching phase so that its computational complexity scales with the vector length rather than the full matrix size, effectively removing the dominant bottleneck of prior designs. Moreover, in the 100K subset of the LAION (Schuhmann et al., 2022) dataset, Compass (Zhu et al., 2024), an Oblivious RAM–based search method, required 76 ms on the fast network and 931 ms on the slow network. In comparison, our encrypted database achieved 62 ms and 251 ms, yielding 1.2× and 3.7× speed-ups, respectively.

| Deep1B | 1K | 4K | 10K | 100K | 1M |
|---|---|---|---|---|---|
| CHAM (Ren et al., 2023) | 378 ms | 389 ms | 1,171 ms | 9,406 ms | 84,543 ms |
| Ours | 150 ms | 151 ms | 156 ms | 236 ms | 951 ms |

Table 4: Single-thread latency of homomorphically encrypted matrix–vector multiplication as the number of database elements increases. The CHAM baseline is based on our re-implementation of the original method, incorporating additional optimizations such as ring packing and packing multiple vectors into a single ciphertext. *Takeaway: Our method achieves up to **88×** speed-up over CHAM, enabling real-time encrypted search at million-scale.*

**Encrypted storage incurs < 5.8× overhead.** Storing high-dimensional vectors in homomorphic ciphertexts introduces nontrivial storage overhead. However, as detailed in Section 3 and Appendix C, our implementation adopts optimizations such as packing multiple vector components into a single ciphertext and omitting unused polynomial coefficients, effectively reducing space requirements. Moreover, we apply module-LWE variants and seed-based ciphertext generation techniques, which scale ciphertext size *linearly* with vector dimensionality rather than polynomial degree. As a result, the encrypted database achieves practical storage costs, less than **5.8×** overhead even for millions of entries, enabling deployment in real-world systems without requiring excessive disk resources.

## 5 Conclusion and Discussion

We present a privacy-preserving framework that partitions tasks between powerful but untrusted remote models and lightweight trusted local models. Our key contributions enable secure collaboration without revealing sensitive data. By integrating Socratic-CoT, our framework not only safeguards privacy but also sustains strong performance. In addition, our homomorphically encrypted vector database supports million-scale collections with sub-second latency and negligible accuracy loss compared to plaintext search. Taken together, this work demonstrates that strong privacy guarantees can be achieved without sacrificing performance or scalability, providing a practical foundation for the secure real-world deployment of large-scale AI systems.

Future work should extend our framework by incorporating query protection techniques, as user queries themselves may also contain sensitive information. Since our approach is orthogonal to data minimization and sanitization, combining them can strengthen both query and context privacy.

## 6    REPRODUCIBILITY STATEMENT

We have made extensive efforts to ensure the reproducibility of our work. The main paper provides a detailed description of our proposed privacy-preserving framework (Sections 2– 3), including the design of the Socratic-CoT and the Homomorphically Encrypted Vector Database. Appendix C contains full mathematical derivations, algorithmic details, and additional explanations of the database operations. Appendix D describes the full experimental setup, including benchmark choices, hyperparameter settings, and evaluation metrics. Appendix E specifies the compute resources used, and the exact networking conditions emulated during evaluation. Finally, Appendix H lists all prompt templates used for sub-query generation, chain-of-thought reasoning, and response generation. An anonymized source code and scripts is also submitted as supplementary material to facilitate reproduction of all reported experiments.

## 7    ETHICS STATEMENT

We acknowledge and adhere to the ICLR Code of Ethics[3]. Our study does not involve human subjects or demographic data, but directly addresses critical ethical concerns of privacy and security in large language model interactions. By introducing a framework that separates trusted and untrusted zones and enabling encrypted semantic search over private data, our work strengthens user privacy while maintaining strong utility. We believe this contributes positively to ethical AI development by reducing risks of data exposure, insider threats, and misuse, and by providing a practical foundation for deploying powerful AI systems without compromising personal privacy.

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

## A    RELATED WORK

In this paper, we address the challenge of privacy-preserving LLM interaction, focusing on protecting user records in the context, at inference time. Unlike private training approaches which safeguard the training corpus through techniques like DP-SGD and DP-ICL (Abadi et al., 2016; Tang et al., 2023), we focus on protecting user data provided as inputs to the model during inference, ensuring that sensitive context information remains confidential and is not leaked or memorized by the remote LLMs. Our work intersects with the following topics:

**Private Inference via Encryption.** Early approaches combined homomorphic encryption (HE) with neural networks, exemplified by CryptoNets (Gilad-Bachrach et al., 2016), though with $10^3\times$ computational overhead. Subsequent systems like Gazelle (Juvekar et al., 2018) and XONN (Riazi et al., 2019) reduced latency by hybridizing HE with garbled circuits and binary networks. Recent work extends these techniques to Transformers and LLMs: MPCFormer (Li et al., 2022), PermLLM (Zheng et al., 2024), and PUMA (Dong et al., 2023) achieve privacy for BERT and LLaMA architectures but still require seconds per token. Industry implementations like Apple's HE+PIR photo search (Inc.) show promise, but cloud LLM providers have been reluctant to adopt these approaches due to significant computational overhead and complex key management.

**Input Minimization and Sanitization Methods.** Complementary approaches focus on sanitizing prompts before transmission. PREEMPT (Chowdhury et al., 2025) detects and replaces sensitive spans with placeholders or differentially private values. PAPILLON (Siyan et al., 2024) divides processing between local lightweight models and external LLMs, sending only abstracted prompts to the cloud. Additional work Dou et al. (2023); Staab et al. (2024) focuses on abstracting personal information. While effective for specific domains, these approaches typically require task-specific engineering or sacrifice accuracy when critical context is removed Xin et al. (2025). Our framework preserves task performance without cryptographic overhead by keeping raw data in the trusted zone while delegating only non-sensitive reasoning steps.

**Chain-of-Thought Reasoning and Task Decomposition.** Chain-of-thought (CoT) prompting has emerged as a powerful technique for improving LLM reasoning through step-by-step solutions. Zero-shot CoT techniques (Wei et al., 2022; Kojima et al., 2022) and task decomposition prompts (Zhou et al., 2022; Press et al., 2022) guide models to break complex problems into manageable sub-problems, often enhanced with supervised reasoning traces (Zelikman et al., 2022). Parallel work on model cascades aims to maximize efficiency by routing queries between different-sized models, as in FrugalGPT (Chen et al., 2023) and Hybrid LLM (Ding et al., 2024), typically using confidence estimators to determine when smaller models are insufficient (Liu et al., 2024; Gupta et al., 2024). Multi-model frameworks like Socratic Models (Zeng et al., 2022) and HuggingGPT (Shen et al., 2023) divide tasks between a powerful LLM planner and specialized executors, but assume the central model has full access to private data. In contrast, our approach performs test-time CoT decomposition without additional training while preserving privacy by ensuring the large LLM only sees abstracted queries rather than raw private data.

**RAG and Agentic Workflows.** Recent systems increasingly embed LLMs within persistent user-centric datastores to deliver personalized assistance. These range from research prototypes like Generative Agents (Park et al., 2023) that maintain interaction histories as long-term memory, to commercial deployments such as ChatGPT's "Memory" and Operator (OpenAI, 2024; 2025) that preserve multi-day conversation logs, and open frameworks like LangChain and LlamaIndex (Mavroudis, 2024; LlamaIndex Team, 2024) that provide memory backends as first-class primitives. Life-logging assistants like Rewind and Lindy (Rewind AI, 2024; Lindy AI, 2024) index users' entire digital traces, leveraging the success of retrieval-augmented generation (RAG) (Lewis et al., 2020a) for grounding LLMs in external knowledge. However, these systems typically assume trustworthy datastores, ignoring privacy risks highlighted by recent extraction and inference attacks (Bianchi et al., 2023). Our framework is the first to combine an agentic RAG architecture with encrypted, local retrieval, addressing this critical privacy gap while maintaining the benefits of contextual personalization.

## B    STORAGE LIMITATION ON EDGE DEVICES

We estimate the storage requirements when accumulating long-term user data locally on a personal device. A single data record with its 768-dimensional `float32` embedding requires approximately 103 KB (3 KB for the key embedding and 100 KB for the data value). Thus, 1M records per user

would consume around 103 GB in total, corresponding to more than 80% of the 128 GB storage available on a base iPhone 16. Allocating such a large portion of disk capacity for a single application is clearly impractical, especially as user data continues to grow.

## C DETAILS ON THE HOMOMORPHICALLY ENCRYPTED VECTOR DATABASE

### C.1 MATHEMATICAL NOTATIONS

Given a power-of-two integer $d > 1$, let $\mathcal{R}_{*,d} = \mathbb{Z}[X]/(X^d + 1)$. Given an integer $q > 0$, let $\mathcal{R}_{q,d} = \mathbb{Z}_q[X]/(X^d + 1) \simeq \mathcal{R}_{*,d}/q\mathcal{R}_{*,d}$. Polynomials are written in roman (e.g. q, k) and vectors are written in bold (e.g. $\mathbf{q}, \mathbf{k}$). Given a vector $\mathbf{v} \in F^t$, $v[i]$ denotes the $i$-th coordinate. Given polynomials $p, p' \in \mathcal{R}_{*,d}$, $p \cdot p' \in \mathcal{R}_{*,d}$ denotes the ring multiplication in $\mathcal{R}$. Given a polynomial $p \in \mathcal{R}_{*,d}$, $p[i]$ denotes the coefficient of $X^i$. Given a polynomial in $p \in \mathcal{R}_{*,d'}$, we denote $\tilde{p} \in \mathcal{R}_{*,d}$ as the natural embedding $\tilde{p}(X) = p(X^{d/d'})$. As we use $d$ as the fixed RLWE dimension, we omit $d$ in the notation $\tilde{p}$. For simplicity, we first solve the case where $n = d$ and $r$ is a power of two. For $n \geq d$, we can extend the base case to compute multiple similarity scores. We describe the behavior of the underlying plaintexts.

### C.2 SECURE INNER PRODUCT, ALGORITHMS AND OPTIMIZATIONS

We specify the detailed algorithms as follows. Algorithms 6 and 7 describe the precomputations for the query and key, respectively, as mentioned right after Equation (5). Algorithm 8 describes the score computation algorithm starting from the precomputed query and cache ciphertexts.

**Optimizations Summary.** We summarize the optimizations mentioned in the previous subsection and discuss some additional optimizations.

- **Batching and Caching**: We write the homomorphic inner product equation as in Equation (5). This separates the precomputations for query and key, which are denoted as `Decompose` and `Cache`, respectively. This reduces the number of automorphisms from $d \log(r)$ to $r - 1$.

- **Butterfly Decomposition**: The key side precomputation is significant as it involves $O(r^2)$ polynomial additions. We leverage the butterfly decomposition to reduce the complexity from $r(r - 1)$ to $r \log(r)$.

- **Seeding and MLWE**: In order to improve the storage size, we use Module LWE (MLWE) (Langlois & Stehlé, 2015) and Extendable Output-format Function (XOF) with a public seed. This reduces ciphertext size from $2d$ (i.e. two $\mathcal{R}_{q,d}$ elements) to $r$ (i.e. one $\mathcal{R}_{q,r}$ element and a 128-bit public seed).

- **Remove the leading term** $r$: We use the optimization technique introduced in (Chen et al., 2021) that evaluates the trace without the leading term $r$, thereby improving the precision. This technique is applied for Line 2 of Algorithm 6 and Line 3 of Algorithm 7.

- **Hoisting** (Halevi & Shoup, 2018): We adapt the hoisting technique that lazily computes the homomorphic operations to improve efficiency. Our adaptaion is similar to the double hoisting algorithm in (Bossuat et al., 2021). Hoisting appears in the following instances.

  - Line 3 of Algorithm 6: For each index $0 \leq i < s$, `ModUp`$(a_i)$ is computed only once.
  - Line 5,6 of Algorithm 6, Line 13,14 of Algorithm 7: We `ModDown` after summation, reducing the number of `modDown` to $r$ per each $j$.

- **Reducing NTT dimension**: In Line 3,5,6 of Algorithm 6, we utilize dimension $r$ NTT instead of dimension $d$ NTT, reducing the complexity by a factor of $\log(d)/\log(r)$. This is possible because each $\hat{a}_i$ is sparsely embedded into the larger ring $\mathcal{R}_{q,d}$.

---

**Algorithm 6** `Decompose`

---

**Require:** Query (seeded) MLWE ciphertext $(b, \rho)$ that encrypts $q \in \mathcal{R}_{q,r}$ via the secret key $\mathbf{s} = (s_u)_{0 \leq u < s} \in \mathcal{R}_{q,r}^s$. Here $b \in \mathcal{R}_{q,r}$ and $\rho$ is a 128-bit seed string. $\mathtt{swk}_j = (\mathtt{swk}_{j,u})_{0 \leq u < s} \in (\mathcal{R}_{qp,d}^2)^s$ are the RLWE switching keys where $\mathtt{swk}_{j,u}$ switches from $\tilde{s}_u$ to $\varphi_j^{-1}(s')$ where $s' \in \mathcal{R}_{*,d}$ is the target RLWE secret key. Here `GenA` generates the $a$-part of the MLWE ciphertext from the 128-bit seed $\rho$, and `ModUp` and `ModDown` are the typical homomorphic base conversions from $q$ to $qp$ and from $qp$ to $q$.

**Ensure:** RLWE ciphertexts $(ct_j)_{0 \leq j < r}$ that encrypt $\left(\varphi_j(r^{-1} \cdot q)\right)_{0 \leq j < r}$, i.e. polynomial of degree $d$ in $\mathcal{R}_q$ with $X^{2j+1}$ automorphism operations for $0 \leq j < r$.

1: $\mathbf{a} = (a_u)_{0 \leq u < s} \in \mathcal{R}_{q,r}^s \leftarrow \mathtt{GenA}(\rho)$
2: $(b, \mathbf{a}) \leftarrow r^{-1} \cdot (b, \mathbf{a}) \bmod q$
3: $\hat{\mathbf{a}} = (\hat{a}_u)_{0 \leq u < s} \in \mathcal{R}_{qp,r}^s \leftarrow (\mathtt{ModUp}(a_u))_{0 \leq u < s}$
4: **for** $j = 0$ to $r - 1$ **do**
5: $\quad ct_j \in \mathcal{R}_{qp,d}^2 \leftarrow \sum_{u=0}^{s-1}(\hat{a}_i \cdot \mathtt{swk}_{j,u})$
6: $\quad ct_j \leftarrow \mathtt{ModDown}(ct_j)$
7: $\quad ct_j \leftarrow \varphi_j(ct_j + (\tilde{b} \in \mathcal{R}_{q,d},\ 0))$
8: **end for**
9: **return** $(ct_j)_{0 \leq j < r}$ =0

---

**Algorithm 7** `Cache`

---

**Require:** Key (seeded) MLWE ciphertexts $(b_i, \rho_i)$ that encrypts $k_i \in \mathcal{R}_{q,r}$ via the secret key $\mathbf{s} = (s_u)_{0 \leq u < s} \in \mathcal{R}_{q,r}^s$, for each $0 \leq i < d$. Here $b_i \in \mathcal{R}_{q,r}$ and $\rho_i$ is a 128-bit seed string. $\mathtt{swk}_j = (\mathtt{swk}_{j,u})_{0 \leq u < s} \in (\mathcal{R}_{qp,d}^2)^s$ are the RLWE switching keys where $\mathtt{swk}_{j,u}$ switches from $\varphi_j(\tilde{s}_i)$ to $s'$ where $s' \in \mathcal{R}_{*,d}$ is the target RLWE secret key. Here `GenA` generates the $a$-part of the MLWE ciphertext from the 128-bit seed $\rho$, and `ModUp` and `ModDown` are the typical homomorphic base conversions from $q$ to $qp$ and vice versa, respectively. Let $\mathbf{B} \in \mathcal{R}_{q,d}^{r \times r}$ be the matrix as defined in Equation 7.

**Ensure:** RLWE ciphertexts $(ct_j''')_{0 \leq j < r} \in (\mathcal{R}_{q,d}^2)^r$ that encrypt $\left(\sum_{i=0}^{d-1} \varphi_j(\tilde{k}_i) X^i\right)_{0 \leq j < r}$.

1: **for** $i = 0$ to $d - 1$ **do**
2: $\quad \mathbf{a}_i = (a_{i,u})_{0 \leq u < s} \in \mathcal{R}_{q,r}^s \leftarrow \mathtt{GenA}(\rho_i)$
3: $\quad (b_i, \mathbf{a}_i) \leftarrow r^{-1} \cdot (b_i, \mathbf{a}_i) \bmod q$
4: **end for**
5: **for** $j = 0$ to $r - 1$ **do**
6: $\quad (b_j', \mathbf{a}_j') \in \mathcal{R}_{q,d}^{s+1} \leftarrow \left(\sum_{v=0}^{s-1} \tilde{b}_{v+sj} \cdot X^v, \left(\sum_{v=0}^{s-1} \tilde{a}_{(v+sj),u} \cdot X^v\right)_{0 \leq u < s}\right)$
7: **end for**
8: $\mathbf{ct}' \in (\mathcal{R}_{q,d}^{s+1})^r \leftarrow (b_j', \mathbf{a}_j')_{0 \leq j < r}$
9: $\mathbf{ct}' \in (\mathcal{R}_{q,d}^{s+1})^r \leftarrow \mathbf{B} \cdot \mathbf{ct}'$
10: **for** $j = 0$ to $r - 1$ **do**
11: $\quad ct_j'' = (b_j'', \mathbf{a}_j'') \in \mathcal{R}_{q,d} \times \mathcal{R}_{q,d}^s \leftarrow \varphi_{j,r}(\mathbf{ct}'[j])$
12: $\quad \hat{\mathbf{a}}_j'' = (\hat{a}_{j,u}'')_{0 \leq u < s} \in \mathcal{R}_{qp,d}^s \leftarrow \mathtt{ModUp}(\mathbf{a}_j'')$
13: $\quad ct_j''' \in \mathcal{R}_{qp,d}^2 \leftarrow \sum_{u=0}^{s-1}(\hat{a}_{j,u}'' \cdot \mathtt{swk}_{j,u})$
14: $\quad ct_j''' \in \mathcal{R}_{q,d}^2 \leftarrow \mathtt{ModDown}(ct_j''')$
15: $\quad ct_j''' \leftarrow ct_j''' + (b_j'' \in \mathcal{R}_{q,d},\ 0)$
16: $\quad ct_j''' \leftarrow r \cdot ct_j''' \bmod q$
17: **end for**
18: **return** $(ct_j''')_{0 \leq j < r}$ =0

**Algorithm 8** `Score`

---

**Require:** `Decomposed` query ciphertexts $\mathbf{ct}_q \in (\mathcal{R}_{q,d}^2)^r$, `Cached` key ciphertexts $\mathbf{ct}_k \in (\mathcal{R}_{q,d}^2)^r$.

**Ensure:** A RLWE ciphertext $ct_{out}$ encrypting the resulting score polynomial $\sum_{j=0}^{d-1} \sigma_j X^j$.

  1: $ct_{out} \leftarrow \texttt{Relin}(\sum_{i=0}^{r-1} \mathbf{ct}_q[i] \otimes \mathbf{ct}_k[i])$

  2: **return** $ct_{out}$ =0

---

### C.3 PRIVATE INFORMATION RETRIEVAL

We extend our Secure Inner Product method to support Private Information Retrieval (PIR). Similar to SPIRAL (Menon & Wu, 2022), we treat the database as a matrix. The protocol requires the client to send two encrypted queries: one selecting the target row and the other selecting the target column, each containing a one hot vector at the corresponding index. The server then performs PIR through two sequential applications of the Secure Inner Product protocol. However, naively applying the Secure Inner Product protocol in this PIR context introduces a cache invalidation issue. Specifically, while the standalone Secure Inner Product scenario only requires refreshing the cache corresponding to the updated index, PIR necessitates refreshing the entire cache whenever the database changes. This occurs because the output from the first stage acts as the key for the second stage. To address this, we modify our protocol by applying the inverse butterfly operation—originally intended for use on the key—to the decomposed query instead.

In our experimental setting using a Fast network (see Section E), the modified PIR protocol achieves an end-to-end retrieval latency of under 700 ms for databases consisting of $2^{20}$ records, each sized at 1 KiB. Consequently, we demonstrate that our approach efficiently supports a secure vector database of 1 GiB containing 1 million records with 96 dimensions each, achieving an end-to-end latency below 1 second.

### C.4 NAIVE BASELINE FOR ENCRYPTED DATABASES

Let us consider a simple yet inefficient baseline protocol for implementing a remote encrypted vector database: whenever a client needs to search or update an entry, it downloads the entire database from the server, decrypts it locally, performs the necessary operations, re-encrypts the entire database, and uploads it back to the server. Although straightforward, this approach incurs significant communication overhead and computational burden on the client, making it impractical for large-scale applications. To address these inefficiencies, we aim to design a remote vector database that preserves the same robust security guarantees as the naive approach—where the database remains encrypted under a symmetric key held exclusively by the client—while achieving much greater efficiency in both communication and client-side computation.

### C.5 DETAILED ILLUSTRATION OF HOMOMORPHICALLY ENCRYPTED VECTOR DATABASE

As shown in Figure 3, our system demonstrates how homomorphic encryption enables privacy-preserving retrieval across past and current interactions. In a previous interaction, the user provides sensitive information with a document attachment (e.g., "I have a peanut allergy. Here is my prescription."), which is converted into embeddings on the edge device. The chat history and document attachment are AES-encrypted, while the embedding vectors are homomorphically encrypted and both are sent to the server for secure storage as key–value pairs. At a later time, when the user asks a new question (e.g., "Could you suggest a dinner menu?"), the query is similarly transformed into embeddings, homomorphically encrypted, and transmitted to the server. The server computes similarity scores using ciphertext–ciphertext matrix–vector multiplication, ensuring that search is performed without exposing plaintext data. Since performing top-$k$ selection directly under homomorphic encryption would be prohibitively expensive, the encrypted scores are returned to the client, where they are decrypted and used to perform local top-$k$ selection. Finally, the client retrieves the corresponding records through private information retrieval, decrypts them, and integrates the results into the retrieval-augmented generation process, allowing the model to provide a personalized output while preserving data privacy.

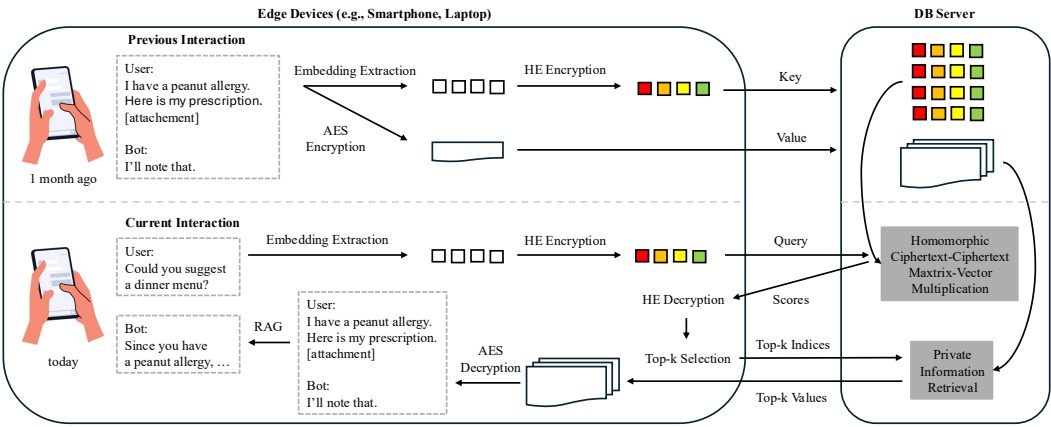

Figure 3: A detailed illustration of the homomorphically encrypted vector database.

# D EXPERIMENTAL SETUP

## D.1 PRIVACY-PRESERVING FRAMEWORK WITH SOCRATIC-COT

We empirically evaluate the effectiveness of our reasoning framework in addressing the computational limitations of local models. Experiments are conducted on two QA-focused benchmarks: LoCoMo, which simulates personal assistant scenarios, and MediQ, which simulates medical consultation scenarios. Both tasks require retrieving relevant private user data and performing complex reasoning to arrive at a final answer. We compare our framework against two categories of baselines: Golden Baselines assume no privacy constraints, allowing private data to be directly passed to remote models. We use GPT-4o (R1), Gemini-1.5-Pro (R2), and Claude-3.5-Sonnet (R3), which cannot be run locally but offer strong reasoning capabilities. Local-only Baselines assume strong privacy constraints, requiring the entire inference process to be carried out by local models. We use Llama-3.2-1B (L1), Llama-3.2-3B (L2), and Llama-3.1-8B (L3), which are lightweight enough for local execution but less capable in complex reasoning tasks. The goal of our reasoning framework is to improve the performance of local-only baselines by leveraging model collaboration and delegated reasoning, aiming to approach the performance of the golden baselines. We use DRAGON (Lin et al., 2023) to extract embedding vectors.

## D.2 HOMOMORPHICALLY ENCRYPTED VECTOR DATABASE

We examine whether vector search can be performed accurately and efficiently over encrypted data using homomorphic encryption. Our goal is to match the quality and latency of plaintext vector search while ensuring that both queries and database contents remain private. The encrypted vector database is implemented using HEXL (Boemer et al., 2021) and evaluated in in the same Google Cloud Platform configuration used by Compass (Zhu et al., 2024) for a fair comparison: an n2-standard-8 instance (8 vCPUs @ 2.8 GHz, 32 GB RAM) as the client and an n2-highmem-64 instance (64 vCPUs @ 2.8 GHz, 512 GB RAM) as the server, co-located in the same region/zone. Using Linux Traffic Control, we emulate two network regimes: Fast (3 Gbps, 1 ms Round Trip Time (RTT)) and Slow (400 Mbps, 80 ms RTT) to isolate the impact of bandwidth and latency. We use 10k query vectors and 1M key vectors from Deep1B (96D) and LAION (512D), as well as the entire LoCoMo dataset (768D). For search accuracy, we report mean/max inner product error, MRR@10, and 1-Recall@k. For latency, we measure end-to-end CPU runtime. All speed measurements assume that both the query and the keys are ciphertexts and employ parameters that satisfy IND-CPA 128-bit security. To evaluate storage, we analyze ciphertext overhead and apply packing optimizations.

## D.3 HYPERPARAMETER SELECTION

To evaluate Socratic-CoT, we set the temperature of all language models to zero to ensure reproducibility. We use top-k retrieval with reranking based on vector similarity scores. We set $k$ to 5 for LoCoMo and 20 for MediQ, as the maximum number of ground truth retrievals varies across datasets.

### D.4 MODEL SELECTION

We employ DRAGON (Lin et al., 2023) as the retriever because it outperforms other candidates, such as DPR (Karpukhin et al., 2020), Contriever (Izacard et al., 2021), and Instructor (Su et al., 2022), on our chosen datasets. It represents data as 768-dimensional vectors, and the inner product between two vectors is used to compute the similarity score. For the remote models, we use GPT-4o (R1) (Hurst et al., 2024), Gemini-1.5-Pro (R2) (Team et al., 2024), and Claude-3.5-Sonnet (R3) (Anthropic, 2024), representing the most powerful closed API language models currently available. These models are assumed to run in a public cloud environment. For the local models, we select Llama-3.2-1B (L1), Llama-3.2-3B (L2), and Llama-3.1-8B (L3) (Dubey et al., 2024), which are lightweight enough to be deployed on edge devices. These models reflect realistic constraints for privacy-preserving, on-device inference. This selection enables a clear evaluation of our framework, balancing reasoning capability with privacy constraints.

### D.5 BENCHMARK SELECTION

We report the performance of Socratic-CoT on two benchmarks. The first, LoCoMo (Maharana et al., 2024), is a benchmark designed to test language models in long-term dialogues. It simulates an everyday personal assistance scenario, where personal information is gradually accumulated in a vector database through extended observation. On LoCoMo, we evaluate (1) the remote models's impact on retrieval using Recall@5 and (2) its enhancement of response quality through improved response generation, measured by the F1 score. We use only the single-hop QA and multi-hop QA datasets out of the total five datasets in LoCoMo, as these are the only datasets suitable for our scenario. The second benchmark, MediQ (Li et al., 2024a), presents a more specialized scenario focused on medical consultation, where privacy risks are directly at odds with the need for access to a patient's personal context. MediQ is a multiple-choice question-answering dataset, so we evaluate generation accuracy using the exact match metric. Since MediQ lacks retrieval annotations, we do not report retrieval metric for this benchmark.

We report the performance of the homomorphically encrypted vector database on standard retrieval benchmarks. To assess the scalability of encrypted storage and search, we selected a sufficiently large dataset. We used the top 10k query vectors and 1M key vectors from Deep1B (Babenko & Lempitsky, 2016) and LAION (Schuhmann et al., 2022), represented as 96-dimensional and 512-dimensional vectors respectively. For LoCoMo (Maharana et al., 2024), we used the entire dataset, which consists of 1,742 query vectors and 4,972 key vectors, each represented as a 768-dimensional vector.

### D.6 METRIC SELECTION

For the Socratic-CoT, we focus on measuring the quality of the generated answers. On the LoCoMo benchmark, we report the F1 score, which captures token-level overlap between generated and ground-truth responses in long-context dialogues. On the MediQ benchmark, we report exact match accuracy, as the task involves multiple-choice question answering and requires strict correctness. These metrics enable us to quantify the impact of delegating complex reasoning to powerful remote models while keeping sensitive data within a trusted zone.

For the homomorphically encrypted vector database, we evaluate both search accuracy and latency. To assess search accuracy, we compute the mean error and maximum error between the inner product similarity scores produced by encrypted and plaintext searches. Additionally, we report 1-Recall@1 and 1-Recall@5, which represent the proportion of queries for which the top-1 result from the plaintext database is not recovered in the top-1 or top-5 encrypted results. Lower values for these metrics indicate higher retrieval consistency under encryption. To evaluate latency, we measure the average response time of encrypted search queries. All metrics are reported separately for plaintext and ciphertext queries.

## E COMPUTE RESOURCES

For Socratic-CoT, all experiments were conducted using a single NVIDIA A100 GPU. Language models from the Llama family were accessed via the Fireworks API (Team, 2025), while other closed API models, including those from OpenAI, Gemini, and Claude, were accessed through their respective APIs. Our homomorphically encrypted vector database was implemented using HEXL (Boemer et al., 2021) and evaluated under the same Google Cloud Platform configuration

used by Compass (Zhu et al., 2024) to ensure a fair comparison: an n2-standard-8 instance (8 vCPUs @ 2.8 GHz, 32 GB RAM) was used as the client, and an n2-highmem-64 instance (64 vCPUs @ 2.8 GHz, 512 GB RAM) was used as the server, both co-located in the same region and zone. To emulate realistic networking conditions, we used Linux Traffic Control to simulate two environments: **Fast** (3 Gbps bandwidth, 1 ms round-trip time and **Slow** (400 Mbps bandwidth, 80 ms round-trip time). The following commands were used to apply these network configurations to the server.

**Fast Network**

```
tc qdisc add dev ens4 root netem delay 1ms
tc qdisc add dev ens4 root handle 1: htb default 30
tc class add dev ens4 parent 1: classid 1:1 htb rate 3096mbps
tc class add dev ens4 parent 1: classid 1:2 htb rate 3096mbps
tc filter add dev ens4 protocol ip parent 1:0 prio 1 u32 \
match ip dst $CLIENT_IP flowid 1:1
tc filter add dev ens4 protocol ip parent 1:0 prio 1 u32 \
match ip src $CLIENT_IP flowid 1:2
```

**Slow Network**

```
tc qdisc add dev ens4 root netem delay 80ms
tc qdisc add dev ens4 root handle 1: htb default 30
tc class add dev ens4 parent 1: classid 1:1 htb rate 400mbps
tc class add dev ens4 parent 1: classid 1:2 htb rate 400mbps
tc filter add dev ens4 protocol ip parent 1:0 prio 1 u32 \
match ip dst $CLIENT_IP flowid 1:1
tc filter add dev ens4 protocol ip parent 1:0 prio 1 u32 \
match ip src $CLIENT_IP flowid 1:2
```

# F ABLATION STUDIES ON SOCRATIC-CoT

To better understand the source of performance gains from Socratic-CoT, we conduct two ablation studies on the LoCoMo (Maharana et al., 2024) and MediQ (Li et al., 2024a) datasets.

**Reasoning augmentation leads to substantial performance gains.** Table 5 compares remote-only and local-only baselines, with and without Socratic-CoT. On LoCoMo, all methods benefit from reasoning augmentation: explicitly prompting the model to reason through intermediate steps leads to clear performance gains. For example, the local-only baseline improves from 64.6 to 82.0, a gain of +17.4 percentage points, while the remote-only baseline improves from 80.6 to 92.6, a gain of +12.0 percentage points. These results suggest that reasoning augmentation through Socratic-CoT is key to performance gains on LoCoMo.

| Method | Model | LoCoMo | MediQ |
|---|---|---|---|
| **Remote-Only Baseline** | R1 | 80.6 | **81.8** |
| **Remote-Only Baseline w/ Socratic-CoT** | R1 + R1 | **92.6** | 67.3 |
| **Local-Only Baseline** | L1 | 64.6 | 32.1 |
| **Local-Only Baseline w/ Socratic-CoT** | L1 + L1 | 82.0 | 32.5 |
| **Hybrid Framwork w/ Socratic-CoT** (ours) | L1 + R1 | 87.7 | 59.7 |

Table 5: First ablation study on Socratic-CoT. LocoMo is evaluated by F1 score, while MediQ is evaluated by exact match. R1 is GPT-4o, and L1 is Llama-3.2-1B. *Takeaway: Reasoning augmentation through Socratic-CoT is the primary driver of performance gains.*

**Delegating both sub-queries and chain-of-thought generation to more powerful models is key.** Table 6 highlights two key observations by isolating the contributions of sub-query generation and chain-of-thought generation.

*First, delegating sub-query generation significantly improves retrieval quality.* On LoCoMo, using a smaller model (Llama-3.2-1B) for sub-query generation limits retrieval performance (Recall@5 = 21.8). When this task is handled by a more capable model (GPT-4o), performance nearly doubles to 44.1. This indicates that sub-query generation often requires deeper understanding and reasoning, which smaller models struggle to achieve. Furthermore, using ground-truth retrieval results boosts performance even more, implying that better sub-query generation—closer to the ideal target—can further enhance final answer quality. On MediQ, the amount of private data per user is so limited that most of the relevant records are retrieved even without high-quality sub-queries, reducing the impact of sub-query generation on overall performance.

*Second, delegating chain-of-thought generation improves final response quality.* On LoCoMo, without any chain-of-thought (N/A), the F1 score is 77.8. Incorporating chain-of-thought reasoning from the smaller model raises it to 85.4, and using GPT-4o improves it further to 89.3. These results demonstrate that guiding generation with reasoning augmentation produced by stronger models plays a critical role in achieving high answer quality. Meanwhile, on MediQ, augmenting reasoning without rich domain knowledge from remote models yields only marginal improvements. In this case, the dominant factor is qualified reasoning criteria generated with rich domain knowledge, which powerful remote models provide far more effectively than smaller local models. We provide a more detailed analysis of the MediQ results in Appendix I.

These findings suggest that local-only baselines, even without disclosing queries, are sufficient as effective personal assistants for casual tasks like LoCoMo. In contrast, specialized domains such as MediQ necessitate leveraging the advanced expertise embedded within powerful remote models to deliver high-quality answers. *Therefore, collaborating with remote models becomes essential for users seeking more accurate responses in expert domains.*

| Sub-Query \ CoT | R1 | L1 | N/A |
|---|---|---|---|
| **GT** | 89.3 | 85.4 | 77.8 |
| **R1** (GPT-4o) | 87.7 | 84.7 | 73.9 |
| **L1** (Llama-3.2-1B) | 84.9 | 82.0 | 64.6 |

(a) LoCoMo

| Sub-Query \ CoT | R1 | L1 | N/A |
|---|---|---|---|
| **All** | 60.4 | 32.1 | 31.4 |
| **R1** (GPT-4o) | 59.7 | 31.8 | 33.2 |
| **L1** (Llama-3.2-1B) | 58.6 | 32.5 | 32.0 |

(b) MediQ

Table 6: Second ablation study on Socratic-CoT on the **LoCoMo** and **MediQ** datasets. LocoMo is evaluated by F1 score, while MediQ is evaluated by exact match. Each row corresponds to a different sub-query generation method: For LoCoMo, GT uses ground-truth private data without sub-query generation (Recall@5=**100.0**), R1 uses GPT-4o (Recall@5=**44.1**), and L1 uses Llama-3.2-1B (Recall@5=**21.8**). For Mediq, All setup uses the full user history as input since no retrieval annotation is available, while R1 and L1 follow the same retrieval configuration as in LoCoMo. Each column corresponds to a different chain-of-thought generation method, where N/A indicates that chain-of-thought reasoning is not used. L1 is used for final response generation across all settings. *Takeaway: Delegating both sub-query and chain-of-thought generation to more powerful models is crucial for optimal performance.*

## G  QUALITATIVE ANALYSIS

We present qualitative examples from the LoCoMo and MediQ benchmarks to illustrate how our system improves response quality under strict privacy constraints. By delegating sub-query generation and chain-of-thought reasoning to a powerful remote model, and executing final response generation locally, our framework ensures that sensitive data never leaves the trusted zone while still benefiting from advanced reasoning capabilities.

### G.1  LOCOMO

**User Query.** *"What motivated Caroline to pursue counseling?"*

This query requires linking the user's past personal experiences to her career decisions, as this information is often buried in long conversational histories.

**Sub-Query Generation by Remote Model.** The remote model generated sub-queries such as: *"Has Caroline discussed any impactful personal experiences related to her career?" "Did she mention an interest in counseling in past conversations?"*

These sub-queries were embedded on the local client and used to search the homomorphically encrypted vector database.

**Encrypted Search from Private Records.** The search retrieved a key statement: *"My own journey and the support I got made a huge difference... I saw how counseling and support groups improved my life."*

**Chain-of-Thought Reasoning from Remote Model.** The model suggested this reasoning guideline: *"When personal growth or transformation is attributed to support or counseling, infer a connection between that experience and a career motivation to help others."*

**Response Generation by Local Model.** Using the retrieved memory and the reasoning instruction, the local model generated the following answer: *"Caroline was motivated to pursue counseling because of her own journey and the support she received, particularly through counseling and support groups."*

### G.2 MEDIQ

**User Query.** *"I've been feeling more forgetful lately and have started falling more often. What should I do?"*

This query suggests a combination of cognitive and physical decline, potentially indicating an underlying neurological issue. Proper assessment requires integration of personal medical context and symptom history.

**Sub-Query Generation by Remote Model.** The remote model generated targeted follow-up questions, including: *"Is there any record of short-term memory impairment?" "Have the falls become more frequent or severe over time?" "Are there other neurological symptoms noted in the history?"*

**Encrypted Search from Private Records.** These sub-queries were executed on encrypted medical records, retrieving relevant notes such as: *"I couldn't remember any of the five things the doctor asked me to recall after ten minutes." "I've been falling more often lately, and it feels like it's getting worse."*

**Chain-of-Thought Reasoning from Remote Model.** The remote model provided the following reasoning instruction to the local model: *"When both progressive memory loss and increased frequency of falls are reported, evaluate for possible neurodegenerative conditions and recommend medical assessment."*

**Response Generation by Local Model.** Based on the retrieved data and reasoning instruction, the local model generated the following concise response: *"Parkinson's disease."*

These examples demonstrate that our framework enables local models to generate informed, context-sensitive responses by leveraging powerful remote models for high-level reasoning. Throughout the process, sensitive user data remains local, ensuring strong privacy guarantees while maintaining or even improving response quality.

## H PROMPT TEMPLATES

For sub-query generation in both the baselines and Socratic-CoT, we used the prompt shown in Figure 4. For response generation in the baselines, the prompt in Figure 5 was used. For Socratic-CoT, chain-of-thought generation was performed using the prompt in Figure 6, and response generation used the prompt in Figure 7. The prompts include substitution keys, which are described in Table 7.

| Key | Description | Illustrative Example |
|---|---|---|
| {user_input} | User input | `I have a fever and a cough. What disease do I have?` |
| {options} | Multiple-choice option. Formatted as bulleted list. For open ended questions, this is replaced with `Empty` instead. | `- Common cold`
`- Flu`
`- Strep throat` |
| {personal_context} | List of retrieved personal contexts in descending order of importance, one item on each line. | `In January 30th, user consumed a half gallon of ice cream.`
`User enjoys cold drink, even in winter.`
`User spends most of the time in their place alone.` |
| {personal_context_json} | List of retrieved personal contexts in descending order of importance, as JSON-formatted array of strings. | `[`
`    "In January 30th, user consumed a half gallon of ice cream.",`
`    "User enjoys cold drink, even in winter.",`
`    "User spends most of the time in their place alone."`
`]` |
| {generated_reasoning} | The output of reasoning generation step. | (omitted) |

Table 7: Substitutions for our prompts. Whenever the listed substitution keys appear on our prompt template, they are substituted into the actual values as described on the right side of the table.

```
You are a sub-query generator.

1.  You are given a query and a list of possible options.
2.  Your task is to generate 3 to 5 sub-queries that help
retrieve personal context relevant to answering the query.
3.  Each sub-query should be answerable based on the user's
personal context.
4.  Ensure the sub-queries cover different aspects or
angles of the query.
5.  If the options text says 'Empty,' it means no options
are provided.

Please output the sub-queries one sub-query each line, in
the following format:
"Sub-query 1 here"
"Sub-query 2 here"
"Sub-query 3 here"

Example 1)

## Query
I have a fever and a cough.  What disease do I have?

## Options
Common cold
Flu
Strep throat

### Sub-queries
"Have user visited any countries in Africa recently?"
"Have user eat any cold food recently?"
"Have user been in contact with anyone who has a COVID-19
recently?"

Test Input)

### Query
{user_input}

### Options
{options}

### Sub-queries
```

Figure 4: Prompt used for sub-query generation in both the baselines and Socratic-CoT.

```
You are a question answering model.

1.  You are given a personal context, a query, and a list
of possible options.
2.  Your task is to generate an answer to the query based
on the user's personal context.
3.  You should generate an answer to the query by referring
to the personal context where relevant.
4.  If the options text says 'Empty,' it means no options
are provided.
5.  If the options are not empty, simply output one of
the answers listed in the options without any additional
explanation.
6.  Never output any other explanation.  Just output the
answer.
7.  If option follows a format like '[A] something', then
output something as the answer instead of A.

Test Input)

### Personal Context
{personal_context}

### Question
{user_input}

### Options
{options}

### Answer
```

Figure 5: Prompt used for response generation in the baselines.

```
Your task is to provide good reasoning guide for students.

You are a chain-of-thought generator.
1.  You are given a query and a list of possible options.
2.  Your task is to provide a step-by-step reasoning guide
to help a student answer the query.
3.  The reasoning guide should clearly show your reasoning
process so that the student can easily apply it to their
query.
4.  Analyze the query and write a reasoning guide for the
student to follow.
5.  If there is a lack of information relevant to the query,
you must identify the missing elements as "VARIABLES" and
write the guide on a case-by-case basis.
6.  If the options text says 'Empty,' it means no options
are provided.

Test Input)

### Query
{user_input}

### Options
{options}

### Chain-of-Thought
```

Figure 6: Prompt used for chain-of-thought generation in Socratic-CoT.

```
You are a question answering model.

1.  Your task is to answer the query based on the teacher's
chain-of-thought decision guide, using additional personal
context.
2.  Read the chain-of-thought decision guide carefully.
3.  If the decision guide contains "VARIABLES" that may
affect the outcome, extract them and determine their values
based on the personal context.
4.  Then, follow the decision guide and apply the extracted
variables appropriately to derive the final answer.
5.  The final answer must be preceded by '### Answer', and
your response must end immediately after the answer.
6.  If the options text says 'Empty,' it means no options
are provided.
7.  If the options are not empty, simply output one of
the answers listed in the options without any additional
explanation.
8.  Never output any other explanation.  Just output the
answer.
9.  If option follows a format like '[A] something', then
output something as the answer instead of A.

### Personal Context
{personal_context_json}

### Chain-of-Thought
{cot}

### Query
{user_input}

### Options
{options}

### Answer
```

Figure 7: Prompt used for response generation in Socratic-CoT.

# I   ADDITIONAL MEDIQ ANALYSIS

As shown in Table 5, the Remote-Only Baseline with Socratic-CoT performs worse than the standard Remote-Only Baseline on MediQ. To understand the cause of this drop, we conducted a detailed qualitative analysis of the model's inputs and outputs. As a result, we found that R1 (GPT-4o), when generating chain-of-thought reasoning, often included the most likely answer without considering the user's personal context. As a result, L1 (Llama-3.2-1B) became strongly biased toward this uncontextualized answer and also ignored the user's personal context. To address this issue, we added explicit rules to the prompt—shown in Figure 8—to reduce this bias and re-ran the experiment under this setup only. With this adjustment, performance improved from 67.3 to 77.0, indicating that the bias was partially mitigated.

```
Your task is to provide good reasoning guide for students.

You are a chain-of-thought generator.
1.  You are given a query and a list of possible options.
2.  Your task is to provide a step-by-step reasoning guide
to help a student answer the query.
3.  The reasoning guide should clearly show your reasoning
process so that the student can easily apply it to their
query.
4.  Analyze the query and write a reasoning guide for the
student to follow.
5.  The student may have less domain knowledge than you,
but they have more context about the situation.
6.  If there is a lack of information relevant to the query,
you must identify the missing elements as "VARIABLES" and
write the guide on a case-by-case basis.
7.  Since you don't have full context about the situation,
your goal is not to choose a final answer but to present a
set of possible answers along with the reasoning steps that
could lead to each one.
8.  If the options text says 'Empty,' it means no options
are provided.

Test Input)

### Query
{user_input}

### Options
{options}

### Chain-of-Thought
```

Figure 8: Prompt used for chain-of-thought generation in the additional MediQ analysis.

