# OpenReview forum: "Privacy-Preserving LLM Interaction with Socratic Chain-of-Thought Reasoning and Homomorphically Encrypted Vector Databases"
_ICLR.cc/2026/Conference — Submitted to ICLR 2026_

### Official Review · Reviewer_hfMq · 2025-10-19

**Soundness:** 3
**Presentation:** 2
**Contribution:** 2
**Rating:** 4
**Confidence:** 3

**Summary:**

To balance the trade-off between privacy and performance in scenarios where tasks require personal information, the paper introduces a client–server framework that leverages a powerful LLM on the server side for reasoning and employs homomorphic encryption for secure data retrieval.

**Strengths:**

1. The idea of decomposing the process and leveraging a powerful server to enhance reasoning while preserving privacy is reasonable.
2. The overall structure of the paper is clear and easy to follow.
3. The efficiency evaluation, particularly Figure 2, is meaningful and provides valuable insights.

**Weaknesses:**

1. Although the proposed approach seems to provide strong privacy protection, the paper lacks an empirical privacy evaluation.
2. The performance experiments are insufficient, e.g. Table 1 only includes two columns. Adding more metrics or additional sub-results would make the experiments on performance more convincing.
3. Regarding the method:
  - While I can identify some novelty in the overall framework presented in Section 2, the contributions and originality in Section 3 are not clearly highlighted. The authors seem to build upon multiple existing techniques and use some mathematical formulations to describe the process. I strongly suggest making the novel contributions more explicit and clearly distinguishing what is new compared to existing baselines (e.g., CHAM).
  - The chosen baselines also seem slightly outdated. It would be better to include comparisons with more recent methods if possible.
  - The communication savings appear to rely primarily on caching and butterfly decomposition. If these techniques are not newly proposed by the authors, relevant references should be added to clarify their originality.

**Questions:**

Suggestions or Minor Weaknesses:
1. In Table 1, clearly indicate in the main text that “L1,L2” refers to different local model choices.
2. Explain technical terms such as “SIMD” when they first appear.

---

> ### Author Response · Authors · 2025-12-04
>
> **Summary:** We thank the reviewer for recognizing the clarity of our framework, the reasonableness of our privacy-performance trade-off, and the value of our efficiency evaluation. We appreciate the constructive feedback on highlighting our cryptographic novelties and will incorporate these clarifications in the final version.
>
> **Q1: Although the proposed approach seems to provide strong privacy protection, the paper lacks an empirical privacy evaluation.**
>
> **A1:** We rely on **provable cryptographic security** rather than empirical privacy evaluation. Empirical evaluations (e.g., inversion attacks) are typically necessary for obfuscation-based or statistical privacy methods (like Differential Privacy) where leakage is probabilistic. In contrast, our framework uses **Homomorphic Encryption (CKKS)** and **non-deterministic AES-256**, which provide mathematical **128-bit IND-CPA security** (see Section 3.4).
>
> * **Context Privacy:** Guaranteed by cryptography. The server processes data entirely in the encrypted domain; breaking this protection is equivalent to breaking the underlying cryptographic standards (RLWE/AES).
> * **Query Privacy:** As detailed in Section 2, we offer a flexible trade-off. Users can achieve complete query privacy by using our encrypted database with a local-only model (ciphertext-ciphertext retrieval), or trade query privacy for performance by using the Socratic-CoT with a remote LLM.
>
> **Q2: The performance experiments are insufficient, e.g. Table 1 only includes two columns. Adding more metrics or sub-results would make it more convincing.**
>
> **A2:** We agree on the importance of detailed performance breakdown. We would like to point the reviewer to **Tables 5 and 6** in the main paper, which provide the requested sub-results and ablation studies.
>
> * **Ablation Studies (Table 5 & 6):** We isolate the gains from *Sub-Query Generation* vs. *Chain-of-Thought Reasoning*. These results demonstrate that while sub-query delegation improves retrieval (Recall), the reasoning augmentation is the primary driver of the F1 score improvements (e.g., \+17.4 pp on LoCoMo).
> * **Benchmarks:** We selected LoCoMo and MediQ specifically because they represent the hardest class of privacy-sensitive tasks (long-context personal history and expert medical reasoning). General QA benchmarks (like HotpotQA) rely on public knowledge and do not fit the threat model of protecting *private* user context.
>
> **Q3: The contributions and originality in Section 3 are not clearly highlighted. Explicitly distinguish what is new compared to baselines (e.g., CHAM).**
>
> **A3:** We will revise Section 3 to explicitly claim our **two core cryptographic novelties** that solve bottlenecks present in baselines like CHAM:
>
> 1. **Query Decompose & Key Caching:** Unlike CHAM (Ren et al., 2023), which performs heavy matrix-vector operations online, we restructure the protocol to precompute and cache key vectors. This reduces the **online** homomorphic inner product complexity to match that of a plaintext operation (changing only the arithmetic domain), removing the dominant bottleneck of prior designs.
> 2. **Butterfly Decomposition for Caching:** To make this caching efficient, we introduce a novel butterfly decomposition specifically for the encryption pipeline. This reduces the number of expensive automorphism operations from linear $O(N)$ to constant factors for the online phase.
>
> **Impact:** As shown in Table 4, these contributions allow us to achieve a **88x speed-up** over CHAM (951 ms vs 84,543 ms for 1M entries), enabling the first practical sub-second latency for encrypted million-scale search.

---

> > ### Author Response · Authors · 2025-12-04
> >
> > **Q4: The chosen baselines seem slightly outdated. Include comparisons with more recent methods.**
> >
> > **A4:** Regarding cryptographic baselines, we compare against **CHAM (Ren et al., 2023\)**, which is the current state-of-the-art for homomorphic matrix-vector multiplication supporting updates.
> >
> > * **Vs. Rhombus (He et al., 2024):** Rhombus uses a *plaintext-matrix* $\\times$ *ciphertext-vector* strategy. This is inapplicable to our threat model because the "matrix" in our case is the user's private database, which **must** be encrypted.
> > * **Vs. HERS (Engelsma et al., 2022):** HERS does not support dynamic updates (insertions/deletions) to individual vectors, making it unsuitable for a personal agent database that changes over time. We believe CHAM is the most appropriate SOTA baseline for our specific threat model (encrypted database \+ dynamic updates), and we outperform it by an order of magnitude.
> >
> > **Q5: Communication savings appear to rely primarily on caching and butterfly decomposition. If these are not newly proposed, add references.**
> >
> > **A5:** We confirm that the **application of butterfly decomposition specifically for Homomorphic Encryption Key Caching** is a novel contribution of this paper. While butterfly structures are standard in FFTs, their derivation to minimize *homomorphic key-switching* and enable *Key-Query Decoupling* (Section 3.2) is our unique theoretical contribution. This specific construction allows the server to update encrypted vectors without client interaction while minimizing ciphertext expansion (to just 1.8x, the smallest reported), which is not found in prior work.
> >
> > **Minor Weaknesses:**
> >
> > * **Table 1 labels:** We will clarify "L1, L2" in the caption.
> > * **SIMD:** We will add the definition (Single Instruction, Multiple Data) upon first use.

---

### Official Review · Reviewer_9zkN · 2025-10-31

**Soundness:** 2
**Presentation:** 3
**Contribution:** 2
**Rating:** 4
**Confidence:** 3

**Summary:**

This paper presents a hybrid framework for privacy-preserving large language model interaction that intelligently divides computation between a trusted local device and an untrusted cloud service. The system combines a Socratic Chain-of-Thought mechanism, which lets a powerful external LLM generate reasoning prompts and sub-queries without accessing private data, with a homomorphically encrypted vector database that enables secure semantic search over encrypted records.

**Strengths:**

1.The proposed architecture strikes an excellent balance between computation power and data privacy by splitting tasks between the cloud and local devices. This hybrid perspective itself is conceptually valuable.
2.The finding that a small local model (Llama-1B) guided by GPT-4o can outperform GPT-4o alone is surprising and thought-provoking. It suggests that effective reasoning guidance may sometimes outweigh raw model scale.

**Weaknesses:**

1. The paper's core claims of practicality and sub-second latency on local devices are unsupported. The "local client" used in the experiments was a Google Cloud server with an 8-core CPU and 32GB of RAM (n2-standard-8). This is in no way equivalent to a smartphone or laptop.
2. The paper claims its hybrid model (L1+R1, 87.7 F1) outperforms the GPT-4o baseline (R1, 80.6 F1). This is misleading. The paper's own ablation study (Table 5) shows that when the baseline (R1) uses the same Socratic-CoT prompt (R1+R1), its performance hits 92.6 F1, easily beating the authors' hybrid model. This proves the architecture isn't superior; the prompt is just better.

**Questions:**

1. Can the authors provide actual latency and memory usage data for the client-side operations (local LLM inference, HE decryption, PIR protocol) running on a real edge device (like a smartphone or a standard laptop)?
2. Given the paper's own data (Table 5) shows the R1+R1 baseline (92.6 F1) is significantly better than the L1+R1 hybrid model (87.7 F1), how can the authors still claim their hybrid framework outperforms GPT-4o?

---

> ### Author Response · Authors · 2025-12-04
>
> **Summary:** We thank the reviewer for the insightful comments on our hybrid privacy-preserving framework. We appreciate the recognition of our architectural balance between compute and privacy, and the finding that guided local models can outperform unguided remote ones. We address the concerns regarding client-side practicality and baseline comparisons below.
>
> **Q1: Can the authors provide actual latency and memory usage data for the client-side operations (local LLM inference, HE decryption, PIR protocol) running on a real edge device (like a smartphone or a standard laptop)?**
>
> **A1:** We acknowledge that our experimental client (n2-standard-8, 32GB RAM) is more powerful than a typical smartphone, but the **resource requirements of our client-side operations are well within the capabilities of modern edge devices** (e.g., iPhone 15/16, high-end Androids, standard laptops).
>
> 1. **Memory Constraints:** The primary memory consumer on the client is the local LLM. We use **Llama-3.2-1B**, which is specifically designed for mobile deployment. Quantized Llama-3.2-1B takes only **a few gigabytes**, which fits comfortably within the 8GB–16GB RAM found in modern smartphones and laptops. The cryptographic keys and current query ciphertexts require negligible memory (\<20 MB) compared to the model.
> 2. **Latency & Compute:**
>    * **Sub-second Latency Claim:** In the paper, **sub-second latency** explicitly refers to the **Encrypted Vector Search** (Stage 3\) running on the server (see Figure 2 and Section 4.2), which we optimized to be 88x faster than prior work (CHAM). This is the system's primary bottleneck.
>    * **Client Operations:** Client-side load consists of embedding extraction, encryption/decryption, and text generation.
>      * *Encryption/Decryption:* These are lattice-based operations (MLWE/RLWE). On modern CPUs (ARMv9 with NEON/SVE2 or x86 AVX), encryption and decryption of query/result vectors take **milliseconds**.
>      * *Generation:* Llama-3.2-1B inference on modern mobile NPUs (e.g., Apple Neural Engine, Qualcomm Hexagon) achieves speeds of 15-50+ tokens/second, ensuring interactive performance.
>
> **Conclusion:** While our testbed used a cloud instance for reproducibility and network simulation consistency, the workload is architected specifically for standard edge hardware specifications.
>
> **Q2: Given the paper's own data (Table 5\) shows the R1+R1 baseline (92.6 F1) is significantly better than the L1+R1 hybrid model (87.7 F1), how can the authors still claim their hybrid framework outperforms GPT-4o?**
>
> **A2:** We claim our framework outperforms the **standard GPT-4o usage (R1)** while strictly preserving privacy, not that it beats a privacy-violating upper bound.
>
> 1. **Privacy vs. Upper Bound:** The **R1+R1** baseline (92.6 F1) represents the *Privacy-Violating Upper Bound*. It achieves high performance by sending **unencrypted private data** to the external GPT-4o for reasoning. This explicitly violates our threat model (Section 2.1).
> 2. **The Fair Comparison:** Our claim is based on comparing our privacy-preserving hybrid model (**L1+R1**, 87.7 F1) against the standard, privacy-agnostic baseline (**R1**, 80.6 F1).
>    * **R1 (GPT-4o alone):** Users typically query LLMs without complex local scaffolding. We outperform this by **\+7.1 pp**.
>    * **L1 (Local alone):** Users prioritizing privacy are stuck with weak local models (64.6 F1). We outperform this by **\+23.1 pp**.
> 3. **The Value Proposition:** We do not claim our architecture yields better reasoning than GPT-4o with full data access. Rather, we claim it **bridges the gap** between secure but weak local execution and powerful but insecure cloud execution. We achieve 95% of the performance of the privacy-violating upper bound (87.7 vs 92.6) while maintaining **zero private data exposure** to the untrusted server.

---

### Official Review · Reviewer_ZooK · 2025-10-31

**Soundness:** 3
**Presentation:** 3
**Contribution:** 3
**Rating:** 6
**Confidence:** 4

**Summary:**

This paper proposes a hybrid framework for privacy-preserving LLM-based personal assistants, combining Socratic Chain-of-Thought Reasoning (Socratic-CoT) with a Homomorphically Encrypted Vector Database. The system decomposes user queries, leverages an untrusted LLM for reasoning and sub-query generation (without access to private data), and performs encrypted semantic search over local private data. The final answer is generated by a local LLM using the retrieved records. Experiments on LoCoMo and MediQ benchmarks show the hybrid system outperforms local-only baselines and approaches oracle performance, with ablation studies on Socratic-CoT and encrypted search.

**Strengths:**

⦁	Highly innovative, balancing privacy and practicality.
⦁	Comprehensive experiments, including ablation analysis.
⦁	Strong potential for real-world deployment.

**Weaknesses:**

⦁	Significant computational and communication overhead of HE not fully analyzed.
⦁	Local device burden and user experience not discussed.
⦁	Security analysis and threat model are insufficient.
⦁	COT decomposition robustness and fallback mechanisms are not explored.
⦁	Lack of quantitative privacy-utility tradeoff analysis.
⦁	Limited implementation details and reproducibility.

**Questions:**

⦁	What is the end-to-end latency and throughput of the system under realistic deployment?
⦁	How does the framework perform on resource-constrained devices?
⦁	How are keys managed and how is access pattern leakage prevented in practice?
⦁	How robust is the system to inaccurate or incomplete COT decomposition?
⦁	Can the authors provide more details or open-source code for reproducibility?

---

> ### Author Response · Authors · 2025-12-04
>
> **Summary:** We thank the reviewer for recognizing our framework’s innovation in balancing privacy with practicality and for highlighting our comprehensive experiments. We appreciate the constructive feedback regarding system overhead and robustness, which we address below.
>
> **Q1: What is the end-to-end latency and throughput of the system under realistic deployment?**
>
> **A1:** As detailed in Section 4.2 and Figure 2, our system achieves sub-second latency (627 ms) for searching 1 million entries even under slow network conditions (400 Mbps, 80ms RTT). This practical performance is achieved through two novel cryptographic optimizations:
>
> 1. **Key-Query Decoupling & Caching:** We move heavy polynomial operations offline, reducing online complexity to that of a plaintext inner product (only arithmetic domains differ).
> 2. **Butterfly Decomposition:** This allows efficient cache construction with minimal key-switching. Combined, these techniques yield a **88x speedup** over state-of-the-art HE accelerators like CHAM, enabling real-time throughput suitable for interactive assistants.
>
> **Q2: How does the framework perform on resource-constrained devices?**
>
> **A2:** Our design specifically targets resource-constrained clients (e.g., smartphones) by offloading storage and heavy computation to the cloud.
>
> * **Storage:** Storing 1M embedding records locally would consume \~103 GB (over 80% of a base iPhone's capacity), as shown in Appendix B. Our encrypted cloud database removes this storage limitation.
> * **Computation:** The client only performs lightweight encryption/decryption and runs a small SLM (Llama-3.2-1B). The vector similarity search is executed on the server.
> * **Overhead:** Our optimizations result in a minimal ciphertext expansion ratio (1.8x upload), ensuring the communication burden on the client is negligible compared to the privacy gains.
>
> **Q3: How are keys managed and how is access pattern leakage prevented in practice?**
>
> **A3:**
>
> * **Key Management:** We use a hybrid encryption scheme. Vector embeddings are encrypted using **CKKS** (128-bit security), and data values use non-deterministic **AES-256**. Crucially, **all decryption keys remain exclusively on the client device** within the Trusted Zone; the server never possesses the keys.
> * **Access Patterns:** We protect against access pattern leakage using a single-server **Private Information Retrieval (PIR)** protocol for the "Return" phase. This ensures the server cannot determine which records are retrieved, even based on memory access patterns.
>
> **Q4: How robust is the system to inaccurate or incomplete CoT decomposition?**
>
> **A4:** We address robustness through prompt optimization. In our ablation studies (Appendix I), we observed that early remote models occasionally generated biased CoT reasoning. We mitigated this by refining prompts to enforce "case-by-case" reasoning guides, which improved MediQ performance from 67.3% to 77.0%, as explained in Appendix I. Furthermore, even if CoT decomposition is suboptimal, the local model still benefits from the retrieved private context, providing a baseline level of utility that exceeds non-RAG approaches.
>
> **Q5: Can the authors provide more details or open-source code for reproducibility?**
>
> **A5:** Yes. We have already submitted our full codebase in the supplementary materials for reproducibility, including the homomorphic encryption optimizations, prompt templates, and hardware configurations used in our experiments. We will also expand the appendix to include the explicit parameter sets for our CKKS scheme ($N=4096$, $\\log QP=109$) to ensure full reproducibility of the cryptographic benchmarks.

---

### Official Review · Reviewer_pyFK · 2025-11-03

**Soundness:** 2
**Presentation:** 2
**Contribution:** 1
**Rating:** 2
**Confidence:** 5

**Summary:**

The paper proposes Socratic Chain-of-Thought Reasoning (Socratic-CoT), a hybrid privacy-preserving framework for large language model (LLM) reasoning over sensitive user data. The approach decomposes user queries using an untrusted remote LLM, which produces a chain-of-thought and sub-queries without directly accessing user data. These sub-queries are then embedded and used to perform encrypted semantic search over the user’s private database through a Homomorphically Encrypted (HE) Vector Database. Retrieved records are decrypted locally and passed to a smaller, trusted local LLM (e.g., Llama-3.2-1B) to generate the final contextualized response.
Experiments on the LoCoMo long-context QA benchmark demonstrate up to 7.1 percentage-point improvement over GPT-4o alone, showing that hybrid reasoning between remote and local models can improve accuracy while preserving privacy.

**Strengths:**

Timely and relevant problem: Addresses the crucial trade-off between model capability and user privacy for personal assistants handling sensitive information.
Empirical validation: The LoCoMo experiments show that the hybrid setup can outperform a large remote model, supporting the paper’s core hypothesis.

**Weaknesses:**

Insufficient cryptographic detail:
The cryptographic layer lacks rigor. The paper cites Gentry (2009b) and Brakerski et al. (2014) in the main body but does not specify which homomorphic encryption (HE) scheme is actually implemented, nor its parameters or security level. These references are outdated relative to practical schemes like CKKS.
Missing PIR protocol description:
The same issue applies to the Private Information Retrieval (PIR) component. The paper does not indicate which protocol variant is used.
No formal security model or proofs:
Despite repeated privacy claims, there are no formal definitions or security proofs. It remains unclear what type of privacy guarantee is achieved (semantic security, query unlinkability, etc.) and against what adversarial assumptions.
Further clarification is required.
Ambiguous baselines and fairness of comparison:
The description of baselines is incomplete. It is unclear whether GPT-4o has direct access to private data or is restricted. Without a properly defined remote-only baseline, the comparison risks being unfair, as differences in data access could inflate performance gains.
Limited experimental evaluation:
The evaluation relies on only two datasets, which is not sufficient to substantiate generalization claims. The approach should be validated on more diverse and larger datasets.
Unjustified claim on local model limitations:
The statement that local models preserving privacy lack the computational capacity for complex reasoning is asserted but not justified. The authors should clarify whether this refers to hardware limits, parameter size, or theoretical restrictions introduced by privacy mechanisms.
Unexplained mathematical notation in Section 3.2:
Several mathematical symbols and expressions in Section 3.2 are introduced without proper definitions or explanations. This makes it difficult for readers to follow the derivations and evaluate the correctness of the proposal.

**Questions:**

On local model limitations: Why exactly do local privacy-preserving models lack sufficient reasoning capacity—hardware limits, model scale, or privacy overhead?
On HE details: Which homomorphic encryption scheme is used? Why you do not provide security proof?
On PIR details: What specific PIR protocol is implemented based on existing literature, and what are its performance characteristics?
On baselines: What exactly is the baseline that the proposed method outperforms, and how is fairness ensured?
On dataset diversity: Why only two datasets? Are additional evaluations planned?

---

> ### Author Response · Authors · 2025-12-04
>
> **Summary:** We thank the reviewer for their constructive feedback, particularly regarding the cryptographic specifications and baseline comparisons. We appreciate the opportunity to clarify the security parameters, the choice of baselines, and the rationale behind our experimental design.
>
> **Response to Weaknesses:**
>
> * **Insufficient cryptographic detail:** Regarding the HE scheme implementation details, Section 3.4 mentions that Key vectors are encrypted using **CKKS** (Cheon et al., 2017). Additionally, the security level is described as **128-bit IND-CPA security** in the same section. The references to Gentry (2009b) and Brakerski et al. (2014) in the Section 2 were included to provide conceptual and historical background on fully homomorphic encryption, while the implementation details, such as the ring structure, are provided in Section 3.2 and Appendix C.1.
> * **Missing PIR protocol description:** The PIR protocol implementation is detailed in Appendix C.3. We describe that our approach adopts the SPIRAL framework (Menon & Wu, 2022\) by treating the database as a matrix, but uniquely integrates our novel homomorphic matrix-vector multiplication scheme (proposed in Section 3.2) into the process. Specifically, Appendix C.3 states, "The server then performs PIR through two sequential applications of the Secure Inner Product protocol." We also describe the specific modification—applying the inverse butterfly operation to the decomposed query—to address cache invalidation. We will ensure this technical synergy between our novel multiplication scheme and SPIRAL is highlighted even more explicitly in the camera-ready version.
> * **No formal security model or proofs:** The security model and guarantees are outlined in Section 2.1 and Section 3.4. Section 2.1 defines the threat model, assuming a semi-honest server and network. In Section 3.4, we mention that the system provides 128-bit IND-CPA security through the use of CKKS and non-deterministic AES-256. As our framework utilizes these standard cryptographic primitives, we rely on their established security properties.
> * **Unexplained mathematical notation in Section 3.2:** We appreciate the feedback regarding readability. While Section 3.2 currently references Appendix C.1 for detailed notations with the statement "The detailed mathematical notations used in the following derivations are provided in Appendix C.1", we acknowledge that frequently referring to the appendix can disrupt the reading flow. In the camera-ready version, we will revise the main text to include brief definitions of core mathematical symbols (e.g., ring structures and automorphism operations) directly within Section 3.2, ensuring the derivations are self-contained and easier to follow.

---

> ### Author Response · Authors · 2025-12-04
>
> **Response to Questions:**
>
> **Q1: On local model limitations: Why exactly do local privacy-preserving models lack sufficient reasoning capacity—hardware limits, model scale, or privacy overhead?**
>
> **A1:** The limitation is primarily driven by the hardware constraints of edge devices (e.g., smartphones, laptops) relative to the scale of personal data. Real-world personal data—such as conversation logs and documents—accumulates rapidly, often reaching hundreds of gigabytes, as shown in Appendix B. Storing and indexing this volume locally overwhelms the storage and memory capacity of consumer devices. While local models (like Llama-3.2-1B) are improving, running multi-billion parameter models alongside large-scale vector retrieval locally creates a critical bottleneck. Our framework bridges this by offloading the heavy storage and retrieval to the cloud via HE, while keeping the high-level reasoning process (CoT) on powerful remote models without exposing data.
>
> **Q2: On HE details: Which homomorphic encryption scheme is used? Why you do not provide security proof?**
>
> **A2:** We use the CKKS scheme as stated in Section 3.4. Regarding security proofs, we utilize standard primitives—CKKS for homomorphic encryption and AES-256 for symmetric encryption—which have established security guarantees (IND-CPA) in their respective original papers (Cheon et al., 2017; Rogaway, 2004). Section 3.4 references Bossuat et al. (2024) to indicate that our parameters align with security guidelines.
>
> **Q3: On PIR details: What specific PIR protocol is implemented based on existing literature, and what are its performance characteristics?**
>
> **A3:** Our implementation is based on SPIRAL (Menon & Wu, 2022\) as described in Appendix C.3, with modifications to handle dynamic updates. Performance-wise, Appendix C.3 reports that the protocol achieves an end-to-end retrieval latency of under 700 ms for databases with $2^{20}$ records.
>
> **Q4: On baselines: What exactly is the baseline that the proposed method outperforms, and how is fairness ensured?**
>
> **A4:** We compare against two distinct baselines to define the privacy-utility spectrum: (1) Local-Only (Privacy Bound): Models like Llama-3.2-1B running entirely on-device (maximum privacy, lower capacity). (2) Remote-Only (Utility Bound/Oracle): Models like GPT-4o with direct access to plaintext private data (minimum privacy, maximum capacity). This comparison is fair as it explicitly quantifies the trade-off. Our method outperforms the Local-Only baseline by utilizing remote reasoning (Socratic CoT) and approaches the performance of the Remote-Only "oracle" without compromising privacy. We will clarify these definitions in the revision.
>
> **Q5: On dataset diversity: Why only two datasets? Are additional evaluations planned?**
>
> **A5:** We selected LoCoMo (long-term memory) and MediQ (clinical reasoning) because they specifically require retrieval over private, user-specific data, unlike generic benchmarks (e.g., HotpotQA) which rely on public knowledge. These datasets effectively test the system's ability to handle sensitive personal context, which is our core focus. We acknowledge the need for broader evaluation and are actively running experiments on additional privacy-centric datasets to be included in the final version.

---

### Meta-Review · Area_Chair_wuTe · 2025-12-13

**Summary:**

The paper presents a hybrid method for privacy preserving personal agents based on LLMs. Reviewers appreciated the importance of the problem and the and the hybrid framework as a promising framework for a solution. At the same time, concerns were raised about the scope of the experimental setup, the level of technical detail, and the clarity of novelty and contributions. The rebuttals were comprehensive and have addressed most of these points, in many cases pointing to content that had already been included in the original submission, which makes it difficult to gauge the source of the large number of questions within the unique confines of this particular conference. In light of the pending questions and an overall view of the reviews and rebuttals, it is concluded the paper would benefit from a revision and another comprehensive round of review. The recommendation is therefore to decline it for this conference and encourage a resubmission.

**Reviewer Concerns:**

Many questions were raised about the level of technical detail in the paper. The rebuttals for the most part reiterate that those details had actually been present in the original submission. It is therefore hard to say what was the source of those concerns and whether they have been addressed. Another concern was regarding which parts are novel in this paper compared to prior work. While the rebuttal has addressed that, the overall impression from both kinds of concerns taken together is of a paper somewhat lacking in clarity of presentation.

Another concern raised by some reviewers was the scope of the experimental evaluation. The rebuttal stated that additional experiments were planned for the final version, although they were not included in the rebuttal, so that concern cannot be counted as fully addressed.

**Reviewer Scores:**

It is particularly hard in this case to say how reviewers might have changed their scores, given multiple apparently factual disagreements between the reviews and the rebuttals which did not get a chance to arrive at a resolution. However, considering the totality of scores and reviews, it is unlikely that possible score changes would have changed the outcome for this paper.

---

### Decision · Program_Chairs · 2026-01-26

Reject